# α3 Chains of type V collagen regulate breast tumour growth via glypican-1

Guorui Huang[1], Gaoxiang Ge[1,†], Valerio Izzi[2] & Daniel S. Greenspan[1]

Pericellular α3(V) collagen can affect the functioning of cells, such as adipocytes and pancreatic β cells. Here we show that α3(V) chains are an abundant product of normal mammary gland basal cells, and that α3(V) ablation in a mouse mammary tumour model inhibits mammary tumour progression by reducing the proliferative potential of tumour cells. These effects are shown to be primarily cell autonomous, from loss of α3(V) chains normally produced by tumour cells, in which they affect growth by enhancing the ability of cell surface proteoglycan glypican-1 to act as a co-receptor for FGF2. Thus, a mechanism is presented for microenvironmental influence on tumour growth. α3(V) chains are produced in both basal-like and luminal human breast tumours, and its expression levels are tightly coupled with those of glypican-1 across breast cancer types. Evidence indicates α3(V) chains as potential targets for inhibiting tumour growth and as markers of oncogenic transformation.

[1] Department of Cell and Regenerative Biology, University of Wisconsin School of Medicine and Public Health, Madison, Wisconsin 53705, USA. [2] Centre of Excellence in Cell-Extracellular Matrix Research and Biocenter Oulu, Faculty of Biochemistry and Molecular Medicine, University of Oulu, Aapistie 5, Oulu 92100, Finland. † Present address: Institute of Biochemistry and Cell Biology, Shanghai Institutes for Biological Sciences, Chinese Academy of Sciences, Shanghai 200031, China. Correspondence and requests for materials should be addressed to D.S.G. (email: dsgreens@wisc.edu).

Collagen V [col(V)] is a low abundance fibrillar collagen widely distributed in tissues as $\alpha1(V)_2\alpha2(V)$ heterotrimers[1] that integrate into fibrils of the abundant collagen I [col(I)] and regulate the geometry of resulting col(I)/col(V) heterotypic fibrils[2]. $\alpha1(V)_2\alpha2(V)$ heterotrimers also regulate the tensile strength of col(I)/col(V) fibrils, as mutations in the genes for either the $\alpha1(V)$ or $\alpha2(V)$ chain can cause classic Ehlers–Danlos syndrome[3,4], which is characterized by fragile connective tissues[5]. There is a third col(V) chain, $\alpha3(V)$, which can be found in $\alpha1(V)\alpha2(V)\alpha3(V)$ heterotrimers and has a more limited tissue distribution than do $\alpha1(V)_2\alpha2(V)$ heterotrimers[6]. Tissues in which the $\alpha3(V)$ chain has been detected include white adipose tissue (WAT), skeletal muscle, and pancreatic islets, in which pericellular $\alpha3(V)$ chains are important to proper functioning of adipocytes, myofibres and pancreatic $\beta$ cells, respectively[6]. $\alpha3(V)$ RNA is at relatively high levels in breast[7]. Thus, findings of high $\alpha3(V)$ levels in WAT[6] suggested that high $\alpha3(V)$ levels in breast might occur in mammary fat pads. We show here that $\alpha3(V)$ chains are in mammary fat pads, but are also at particularly high levels in association with, and are produced by, mammary gland basal cells.

Interactions between epithelial cells and the extracellular matrix (ECM) are important to breast carcinoma pathogenesis. Stromal fibrillar collagens seem of particular importance, as their density helps determine breast carcinoma risk, and fibrils can provide tracks along which metastatic epithelial cells migrate[8]. Col(V) is specifically upregulated ∼10-fold in the desmoplasia associated with scirrhous infiltrating ductal carcinomas[9], suggesting a role in breast cancer aetiology.

The importance of collagenous ECM to breast carcinoma etiology, the specific upregulation of col(V) in desmoplasia, and the high $\alpha3(V)$ levels associated with mammary gland prompted us to assess possible $\alpha3(V)$ roles in mammary carcinoma aetiology. Towards this end, effects of ablating the $\alpha3(V)$ gene Col5a3 on mammary tumour biology were studied in the MMTV-PyMT mouse model, which recapitulates many processes observed in human breast cancer progression and metastasis[10].

MMTV-PyMT tumour progression was markedly slowed by $\alpha3(V)$ ablation, predominantly due to tumour cell autonomous effects. $Col5a3^{-/-}$ MMTV-PyMT tumour cells had greatly reduced proliferative potential, apparently due to loss of interactions between $\alpha3(V)$ and the cell surface proteoglycan glypican-1 (GPC1), affecting ability of the latter to act as a co-receptor for mitogenic factors. Results thus provide a previously unrecognized mechanism for control of tumour progression by collagenous ECM. $\alpha3(V)$ chains and GPC1 are shown to be separately expressed by basal and luminal cells, respectively, in normal mouse and human mammary gland, but to be coexpressed in luminal and basal-like human breast tumours, which may provide a 'gain of autonomy' and thus growth advantage to tumour cells. Such growth advantage may be of particular importance to luminal A tumours, with which particularly high $\alpha3(V)$ and GPC1 expression levels are strongly associated. Data showing anti-$\alpha3(V)$ antibodies to slow tumour cell growth in vitro and in vivo suggest avenues for therapeutic interventions.

## Results

### Col5a3 ablation slows tumour growth in MMTV-PyMT mice.
Immunofluorescence found that $\alpha3(V)$ chains, although detected throughout mammary fat pads, are at especially high levels associated with mammary glands (Fig. 1a). In contrast, anti-$\alpha1(V)$ and -$\alpha2(V)$ antibodies showed $\alpha1(V)_2\alpha2(V)$ heterotrimers to be evenly distributed between fat pad and glands, suggesting enrichment of only $\alpha3(V)$-containing col(V) within the latter. Co-localization showed high $\alpha3(V)$ levels of mammary glands to

be exclusively associated with basal cells (Fig. 1b), with no apparent association with luminal cells (Fig. 1c).

To determine possible effects of $\alpha3(V)$ ablation on mammary carcinomas, $Col5a3^{-/-}$ and MMTV-PyMT mice were intercrossed to obtain $Col5a3^{-/-}$/MMTV-PyMT (KO/PyMT) and $Col5a3^{+/+}$/MMTV-PyMT (WT/PyMT) progeny (Supplementary Fig. 1a). Although significant difference in tumour latency between KO/PyMT and WT/PyMT mice was not observed (Supplementary Fig. 1b), KO/PyMT survival was significantly increased (Fig. 1d, $P > 0.0001$, with survival in groups compared using the Log-Rank test) and, at any age, KO/PyMT tumours were markedly smaller than WT/PyMT tumours (Fig. 1e). Thus, tumours grew more slowly and survival was enhanced in KO/PyMT mice. However, despite the reduced size of KO/PyMT mammary tumours, Col5a3 ablation did not appear to significantly affect the extent of lung metastasis (Supplementary Fig. 1c–e).

Although exclusively associated with basal cells in normal mouse mammary ducts (Fig. 1b), $\alpha3(V)$ chains were expressed by WT/PyMT tumour cells (Fig. 1f,g), despite the fact that MMTV-PyMT tumours have gene expression profiles characteristic of luminal type tumours[11]. Indeed, $\alpha3(V)$-positive WT/PyMT tumours were also positive for luminal marker K8 (Fig. 1f), and negative for basal marker K14 (Fig. 1g). In WT/PyMT tumour sections, $\alpha3(V)$ co-localized with K14 only in the basal cells of untransformed ducts (Fig. 1g).

### Cell- and non-cell-autonomous $\alpha3(V)$ effects on tumour growth.
To determine the extent to which decreased tumour growth might be due to stromal, non-tumour cell-autonomous, properties, WT/PyMT primary tumour cells were introduced into the fourth abdominal fat pads of $Col5a3^{+/+}$ (WT) and $Col5a3^{-/-}$ C57BL/6 mice. Survival and tumour sizes of the two groups were then compared at various time points. Survival was markedly increased (Fig. 2a, $P > 0.0001$, with survival in groups compared using the Log-Rank test) and tumour size markedly decreased (Fig. 2b) in $Col5a3^{-/-}$, compared with WT, mice injected with WT/PyMT cells, indicating non-cell-autonomous contributions from microenvironments of stromal origin. To assay for possible tumour cell-autonomous effects of Col5a3 ablation, KO/PyMT primary tumour cells were injected into WT and $Col5a3^{-/-}$ mice. Interestingly, survival of both WT and $Col5a3^{-/-}$ mice injected with KO/PyMT cells was greatly prolonged up to ∼115 days post injection (Fig. 2c) and tumour growth was greatly delayed, with tumours first palpable at ∼40 days post injection (Fig. 2d). These results are in contrast to survival times of WT or $Col5a3^{-/-}$ mice injected with WT/PyMT cells, in which mouse survival was only up to ∼30 or ∼45 days post injection, respectively (Fig. 2a), and in which tumours were first palpable only ∼14 days post injection (Fig. 2b). Superimposition of non-cell-autonomous results of Fig. 2a,b on cell autonomous results (Fig. 2c,d, respectively) illustrates the extents to which cell-autonomous effects exceed non-cell-autonomous effects of Col5a3 ablation on survivability and tumour growth. In fact, survival times of WT versus $Col5a3^{-/-}$ mice injected with KO/PyMT cells were not significantly different from each other ($P < 0.98$, with survival in groups compared using the Log-Rank test), further emphasizing the extent to which cell-autonomous effects exceed non-cell-autonomous effects.

### Decreased KO/PyMT tumour cell proliferative activity.
KO/PyMT tumours have a markedly lower labelling index for proliferation antigen Ki-67 than do WT/PyMT tumours (Fig. 3a,b). This is of interest, as it indicates impairment of

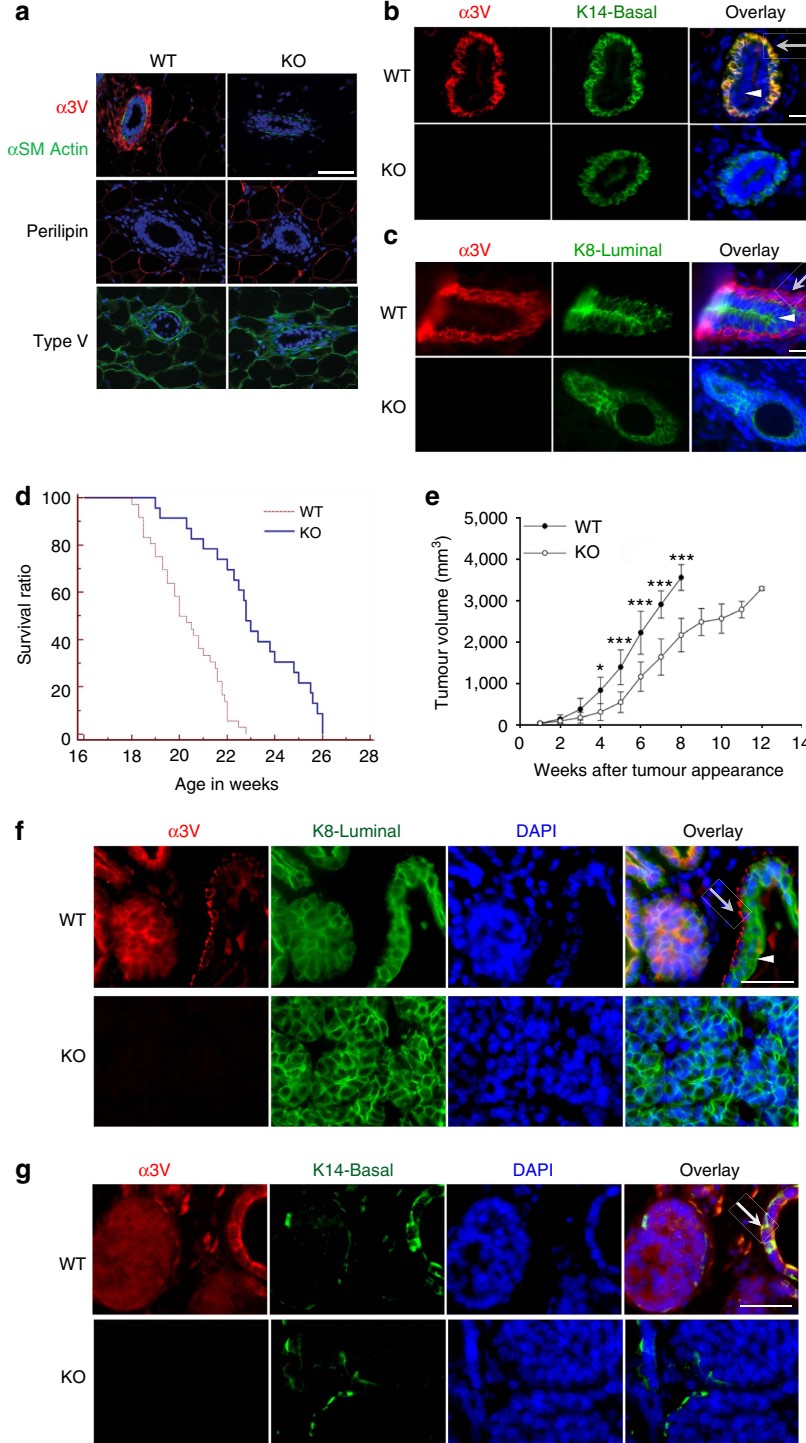

**Figure 1 | Ablation of α3(V), found at high levels juxtaposed to mammary ducts, yields decreased tumour size and increased host survival. (a)** Representative immunofluorescence staining shows α3(V) chains (top panels, red); α-smooth muscle actin (αSM actin, top panels, green), which marks ductal myoepithelial cells; perilipin (adipocyte marker); and col(V) (bottom panels, green). Blue; DAPI staining. Representative immunofluorescence staining shows (**b**) co-localization of α3(V) (red) with marker K14 (green) in basal cells, and (**c**) lack of co-localization of α3(V) (green) with marker K8 (red) in luminal cells, of mammary ducts. Arrowheads and arrows denote luminal and basal cells, respectively. *Col5a3* ablation Kaplan–Meier plots show significantly increased survival ($P < 0.0001$, survival in groups was compared using the Log-Rank test) in KO/PyMT mice lacking a functional *Col5a3* gene (**d**). (**e**) Volume of tumour burden is increasingly reduced, relative to WT/PyMT controls, in KO/PyMT mice at times after initial tumour appearance. All palpable tumours were measured for volume calculations. All WT/PyMT mice were killed by 8 weeks after initial tumour appearance, because of tumour burdens ⩾ 3,000 mm³. (**d,e**) WT/PyMT $n = 34$, KO/PyMT $n = 23$. Data are presented as mean ± s.d. $P$ values: * $< 0.05$, *** $< 0.005$. Statistical analysis was via two-tailed Student's *t*-test, with differences considered significant at $P < 0.05$. Representative immunofluorescent staining shows co-localization of α3(V) (red) with marker K8 (green) (**f**) in the tumour cells of WT/PyMT tumours, and the lack of co-localization with marker K14 (green) in the same tumours, except in basal cells of untransformed ducts (**g**). White scale bars, 50 μm. In **f**, arrowhead and arrow denote luminal and basal cells, respectively, in an untransformed duct. In g, an arrow denotes co-localization of α3(V) and K14 only in the basal cells of an untransformed duct.

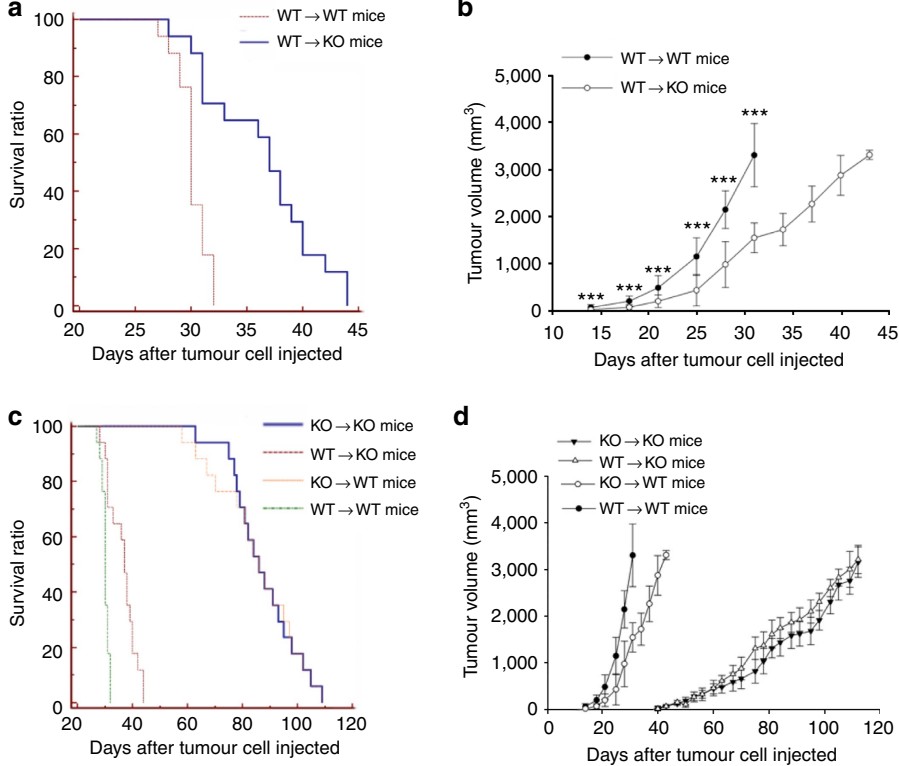

**Figure 2 | Non-cell-autonomous and cell-autonomous effects of *Col5a3* ablation on survivability and tumour volume.** WT/PyMT primary tumour cells were injected into wild type (WT→WT) or *Col5a3*[−/−] (WT→KO) mice (*n* = 8 mice per each group), which were then characterized for survival time (**a**) and tumour volume (**b**) at time points post tumour cell injection. KO/PyMT primary tumour cells were injected into WT (KO→WT) or *Col5a3*[−/−] (KO→KO) mice, which were then characterized for survival time (**c**) and tumour volume (**d**) at time points post tumour cell injection. Injected WT/PyMT cell curves from **a,b** were superimposed on the injected KO/PyMT cell curves in **c,d** to facilitate comparison of survival ratios and tumour volumes. Survival analyses were performed using the Kaplan–Meier method. For tumour volume curves, data are presented as mean ± s.d. *P* value: *** < 0.005. Statistical analysis was via two-tailed Student's *t*-test, with differences considered significant at *P* < 0.05.

KO/PyMT tumour cell proliferation *in vivo*, and as the Ki-67 labelling index can be a prognostic indicator in breast cancers[12]. This difference in Ki-67 labelling index was cell-autonomous, as tumours resulting from KO/PyMT cells injected into WT C57BL/6 mice had much lower Ki-67 labelling indices than did tumours resulting from injection of WT/PyMT cells into WT C57BL/6 mice (Fig. 3c,d).

Consistent with the decreased Ki-67 labelling index, cultured primary tumour cells from KO/PyMT mice had greatly decreased proliferative activity compared with WT/PyMT cells, when assayed via mitochondrial dehydrogenase reduction of MTT (Fig. 3e) or [3]H-thymidine incorporation (Fig. 3f) assays.

In contrast to the marked difference in Ki-67 labelling, no significant difference was found in cleaved caspase 3 levels between KO/PyMT and WT/PyMT tumours or cultured tumour cells (Supplementary Fig. 2). Thus, while decreased proliferative activity likely contributes to reduced KO/PyMT tumour size, differences in apoptotic rates do not.

**Alterations in KO/PyMT tumour cell signalling components.** For insights into intrinsic differences between KO/PyMT and WT/PyMT tumour cells that might affect *in vivo* behaviours, we compared levels and activation of signalling pathway components important to mammary tumour behaviour.

PyMT activates Ras[13], which can signal via its direct effector Raf-1 and the mitogen-activated protein kinase (MAPK) pathway to drive tumour cell growth[14]. We thus compared components of this pathway in KO/PyMT and WT/PyMT tumour cells and tumour tissue samples. Although KO/PyMT and WT/PyMT total

Ras levels did not differ (Fig. 3g,h), levels of activated Ras-GTP, capable of binding Raf-1, were markedly higher in WT/PyMT than in KO/PyMT cell and tumour extracts (Fig. 3h), as were activated phospho-Raf-1 levels (Fig. 3g). Similarly, activated phospho-Erk levels were markedly higher in WT/PyMT than in KO/PyMT samples (Fig. 3g), consistent with increased MAPK signalling downstream of activated Ras-GTP and Raf-1 (ref. 15). Interestingly, phosphorylated FGF receptor (p-FGFR) levels were also higher in WT/PyMT than in KO/PyMT samples, consistent with less signalling by mitogenic factors in the absence of α3(V) chains. All of these differences between WT/PyMT and KO/PyMT samples were significant (Supplementary Fig. 3). Thus, lower MAPK signalling correlates with the intrinsic reduced growth properties of KO/PyMT cells and tumours.

As canonical Wnt signalling levels can affect mammary tumour behaviour via effects on basal stem cell and luminal progenitor populations[16], we compared KO/PyMT and WT/PyMT tumour extracts for β-catenin levels, but found no differences (Fig. 3g).

**KO/PyMT tumour cells have delayed cell cycle progression.** The Ras/Raf/Erk cascade, which has reduced activity in KO/PyMT cells (above), normally transmits mitogenic signals from cell surface receptors (for example, FGFRs) to transcription factors that control cell cycle progression[17]. To obtain further insights into the nature of the decreased proliferative activity of KO/PyMT tumour cells, comparative cell cycle analyses were performed on KO/PyMT and WT/PyMT primary tumour cells. Flow cytometric DNA profiles (Fig. 4a,b) showed that, in normal growth medium, a markedly lower proportion of KO/PyMT than

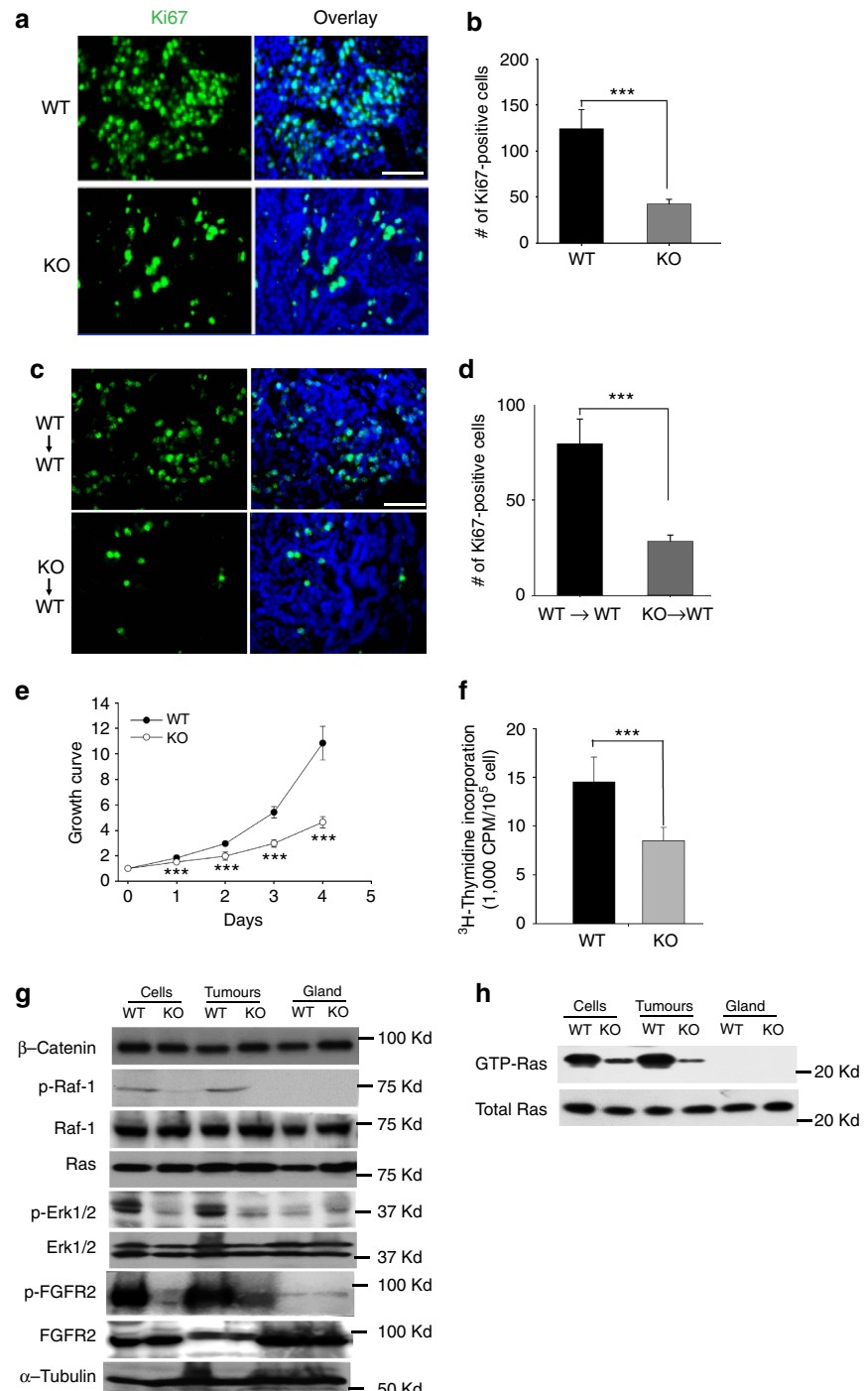

**Figure 3 | KO/PyMT tumour cells have reduced cell-autonomous proliferative activity and Ras/Raf/ERK signaling compared with controls.**
Representative Ki-67 immunohistochemical staining (white scale bars, 50 μm) (**a,c**) and quantification of staining (**b,d**) are shown for WT/PyMT and KO/PyMT tumour sections (**a,b**), and for sections of tumours resulting from injection of WT/PyMT (WT) or KO,PyMT cells into WT or $Col5a3^{-/-}$ mice (**c,d**). Quantification was of six fields per tumour on sections from three different WT/PyMT and three different KO/PyMT tumours, and from three different tumours each from injection of WT/PyMT or KO,PyMT cells into WT or $Col5a3^{-/-}$ mice (WT→WT and KO→WT, respectively). (**e**) Cell proliferation levels of primary KO/PyMT and WT/PyMT tumour cells were assessed via colorimetric MTT assay. Values on the ordinate axis represent OD at 570 nm, given an arbitrary value of 1 at day 0, and then reflecting fold changes on subsequent days. (**f**) $^{3}$H-thymidine incorporation is compared for primary WT/PyMT and KO/PyMT tumour cells. All cell proliferation assays were performed in triplicate. Data are presented as mean ± s.d. P values: **<0.01, ***<0.005. Statistical analysis was via two-tailed Student's t-test, with differences considered significant at P<0.05. (**g**) Representative immunoblots are shown of lysates of WT/PyMT (WT) and KO/PyMT (KO) primary tumour cells and tumours, and of normal C57BL/6 (WT) or $Col5a3^{-/-}$ mammary gland tissue. Blots were probed with antibodies to β-catenin, phospho-Raf-1 (p-Raf-1), Raf-1, Ras, phospho-Erk1/2 (p-Erk1/2), Erk, phospho-FGFR2, FGFR2 or α-tubulin, the latter as a loading control. (**h**) Lysates were subjected to pull-downs with Raf-1 RBD domain to obtain activated Ras (GTP-Ras), which was then detected by immunoblot, using antibody to Ras. An immunoblot displaying total Ras levels in the same samples is also shown. Immunoblots were repeated three times, from three independent tumour or cell lysates. Representative blots are shown. Quantification of results for significance of differences is shown in Supplementary Fig. 3.

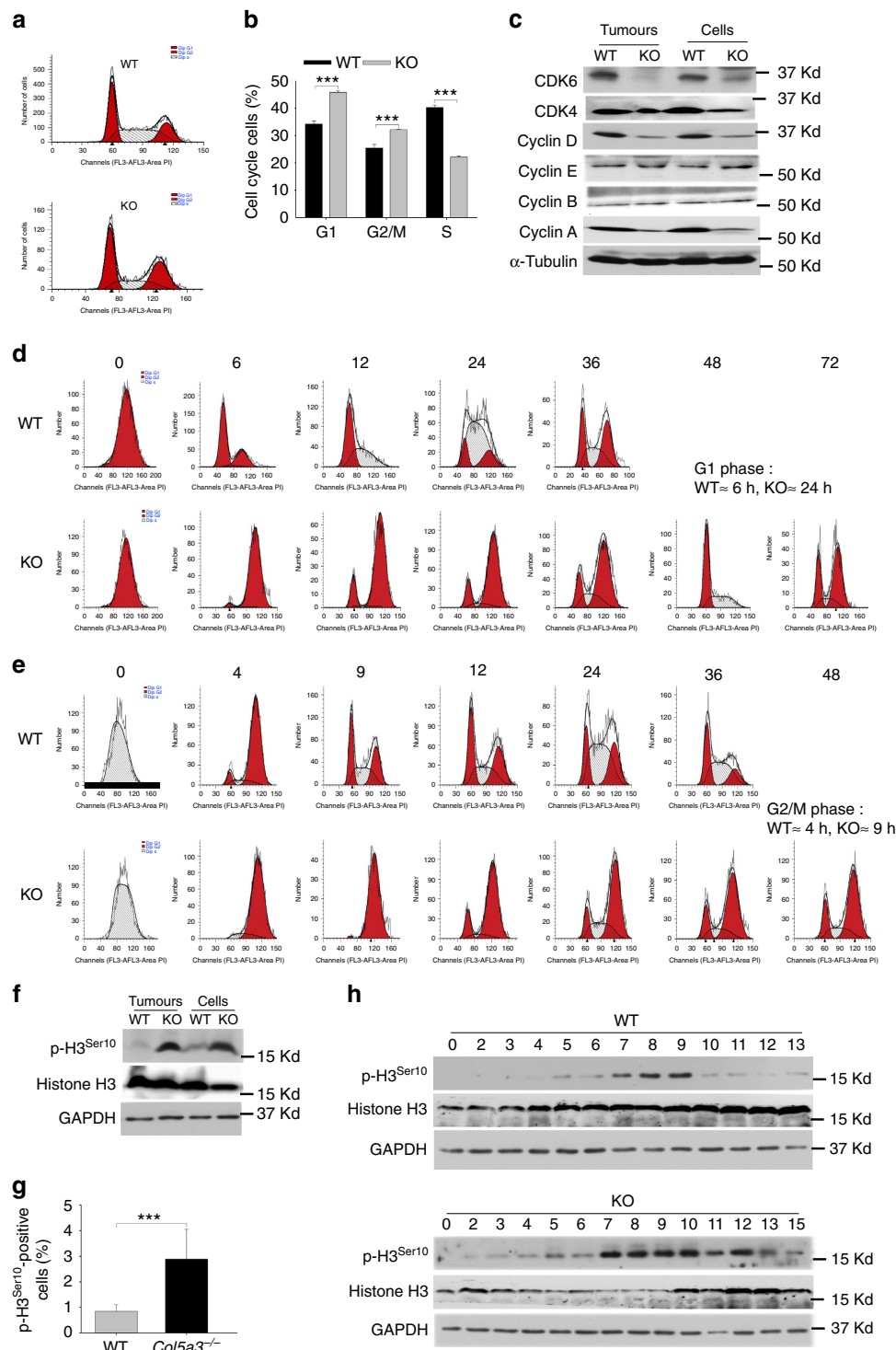

**Figure 4 | Cell cycle delays in KO/PyMT tumours and primary tumour cells.** (**a**) DNA profiles are shown of WT/PyMT and KO/PyMT cells grown in normal growth medium, then stained with propidium iodide and analysed by flow cytometry. (**b**) Quantitation is shown for the DNA profiles of **a**. DNA profiles were performed on triplicate samples. Immunoblotting is shown for cyclins and cyclin regulators (**c**). DNA profiles are shown of cells sorted by FACS into G2M (**d**) and S (**e**) populations, and then released into normal media, harvested at indicated time points (in hours) and analysed by flow cytometry. (**f**) Immunoblotting was done for total histone H3, and antibody to phospho-Serine 10 of histone H3 detected p-H3$^{ser10}$ levels in WT/PyMT (WT) and KO/PyMT (KO) tumour extracts and primary tumour cells. (**g**) Quantification is shown for percent p-H3$^{ser10}$-positive cells detected by immunofluorescent staining of WT/PyMT and KO/PyMT tumour sections. Quantification was of six fields per tumour on sections from three different WT/PyMT and three different KO/PyMT tumours. Data are presented as mean ± s.d. *P* value: *** <0.005. Statistical analysis was via two-tailed Student's *t*-test, with differences considered significant at *P* < 0.05. (**h**) Cells were synchronized at the G1/S border via double thymidine block, released, and then harvested at indicated time points (in hours), and analysed by immunoblot for total histone H3 and p-H3$^{Ser10}$ levels. GAPDH in immunoblots was a loading control. Immunoblots were repeated three times, from three independent tumour or cell lysates. Representative blots are shown. Quantification of results for significance of differences for **c** and **f** is shown in Supplementary Fig. 4.

WT/PyMT cells are in S phase, and markedly larger proportions are in G1 and G2/M, consistent with decreased KO/PyMT proliferative activity.

As cyclin synthesis and consequent cyclin-dependent kinase (CDK) activation are key to regulating cell cycle progression, levels of both were assayed. Lower cyclin D and A levels, but similar cyclin B and E levels, were found in KO/PyMT compared with WT/PyMT cells (Fig. 4c). Lower cyclin D levels are consistent with lower KO/PyMT Ras/Raf/Erk signalling levels (above), as such signalling is thought to affect cell cycle progression by inducing cyclin D expression early in G1 (ref. 17). Lower cyclin A levels may result from the lower proportion of KO/PyMT cells in S phase, in which cyclin A accumulates[18]. Also consistent with lower KO/PyMT proliferative activity were lowered levels of CDK6 and CDK4, which bind and are stabilized/activated by cyclin D[19,20] (Fig. 4c). All these differences between WT/PyMT and KO/PyMT samples were significant (Supplementary Fig. 4a–d). Lower cyclin D-CDK4/6 activity levels indicated by the results suggest partial blockage of KO/PyMT cells in G1, contributing to lowered proliferative activity.

To further assess cell cycle differences, KO/PyMT and WT/PyMT tumour cells were fluorescence-activated cell sorting (FACS) sorted into G1-, S- and G2/M-phase cells, which were then released into growth medium and harvested at indicated time points (Fig. 4d,e, and Supplementary Fig. 4f). By 6 h post sorting, a small number of G2/M-sorted WT/PyMT cells had traversed G1 and were in S-phase (Fig. 4d), showing G0/G1 duration to be ≤6 h. In contrast, for G2/M-sorted KO/PyMT cells, S-phase cells were not clearly detected until 24 h after release into growth medium (Fig. 4d), indicating KO/PyMT G0/G1 duration to be ∼24 h, and thus blockage in G0/G1. To gauge differences in the duration of S and G2/M in WT/PyMT and KO/PyMT cells, S- (Fig. 4e) and G1- (Supplementary Fig. 4f) sorted cells released into growth medium were analysed. For S-sorted cells, WT/PyMT G1-phase cells first appear after 4 h, whereas KO/PyMT G1-phase cells appear, in small numbers, after 9 h, consistent with a delay in G2/M (Fig. 4e). Analysis of G1-sorted KO/PyMT and WT/PyMT cells found no differences in S-phase duration (Supplementary Fig. 4f). Results (summarized in Supplementary Table 1) thus show the KO/PyMT cell cycle to be considerably delayed, with blocks in G1 and G2/M.

As histone H3[Ser10] phosphorylation is crucial for chromosome condensation during mitosis[21], and thus cell cycle progression, it is of interest that KO/PyMT tumours and tumour cells had markedly higher p-H3[Ser10] levels (Fig. 4f and Supplementary Fig. 4e), and that KO/PyMT tumours had a markedly higher proportion of p-H3[Ser10]-positive cells, than did WT/PyMT counterparts (Fig. 4g). H3[ser10] phosphorylation, which begins at prophase, must be removed upon metaphase/anaphase transition for chromosomal decondensation to occur at the end of mitosis[21]. A double thymidine block, to synchronize cells at the G1/S border, followed by analysis of timing for p-H3[ser10] appearance/disappearance upon release of cells from the block, showed highest p-H3[ser10] levels beginning in both KO/PyMT and WT/PyMT cells ∼7 h after G1/S border block release (Fig. 4h). This showed no difference in time needed for KO/PyMT and WT/PyMT cells to make the S/G2 transition, or transverse S before H3[ser10] phosphorylation. However, high p-H3[ser10] levels persisted only 3 h in WT/PyMT cells, but at least 6 h in KO/PyMT cells (Fig. 4h), consistent with delayed dephosphorylation. Thus the KO/PyMT G2/M block evidenced in Fig. 4e is shown by p-H3[ser10] data (Fig. 4h) to be due to prolonged mitosis.

In sum, data are consistent with blocks in G1 and mitosis of KO/PyMT cells, considerably delaying the cell cycle and correlating with lowered proliferative activity.

**α3(V) chains interact with cell surfaces via GPC1.** Important to understanding how α3(V) chains affect tumour cell behaviour is determining the cell surface moieties with which they interact. It has been reported that α4(V) collagen chains, which are similar to, if not orthologous to, α3(V) chains, affect Schwann cell function via interactions with cell surface heparan sulfate proteoglycan (HSPG) GPC1 (ref. 22). As GPC1 is over-expressed in breast and other cancers, in which it appears to regulate the Ras/Erk pathway and modulate mitogenic responses to HSPG-binding growth factors[23–28], we tested for possible α3(V)-GPC1 interactions in WT/PyMT tumours.

Immunofluorescence showed GPC1 to be present in both WT/PyMT and KO/PyMT tumours, and to co-localize with α3(V) in the former (Fig. 5a). Moreover, α3(V) immunoprecipitation from WT/PyMT tumour extracts co-precipitated GPC1 (Fig. 5b) and, conversely, immunoprecipitation of GPC1 co-precipitated α3(V) chains (Fig. 5c). Thus, α3(V) chains co-localize with and are bound to GPC1 in WT/PyMT tumours.

Probing the immunoprecipitates of Fig. 5b,c with antibodies to the α1(V) chain of col(V) showed α1(V) chains to have co-precipitated with α3(V) chains and GPC1 (Fig. 5b,c), consistent with the probability that tumour α3(V) chains bind GPC1 in the context of α1(V)α2(V)α3(V) heterotrimers. Immunoprecipitates were also probed with antibody to the α1(XI) chain of col(XI) as 1) α1(XI) is very similar in structure to α1(V) and capable of substituting for α1(V) in binding other col(V) chains[29–31], such that it was possible that α3(V) chains might occur in α1(XI)-containing heterotrimers; and 2) α1(XI) is a marker that is upregulated in, and may play a role in facilitating various cancers, including breast cancer[32]. However, α1(XI) chains were not detected in α3(V) or GPC1 immunoprecipitates from WT/PyMT tumour lysates (Fig. 5b,c).

We previously suggested that α3(V) might interact with cell surface moieties via highly charged sequences in its N-terminal non-triple helical domain (NTD)[6]. In pull-downs of recombinant proteins, α3(V)-NTD immunoprecipitation was sufficient to co-precipitate GPC1 (Fig. 5d), and vice versa (Fig. 5e). Thus α3(V) NTD sequences are sufficient to bind GPC1.

**GPC1 affects stability/deposition of α3(V)-containing ECM.** To determine the extent to which abrogation of GPC1 binding might affect association of α3(V) chains with WT/PyMT tumour cells, the latter were infected with previously described[33] adenoviral vector (Ad-Sh-1) for shRNA GPC1 knockdown. Interestingly, upon GPC1 knockdown, α3(V) chains were no longer readily detectable by immunofluorescence (Fig. 5f), and were found by immunoblotting to be markedly decreased (Fig. 5g,h). In contrast, α1(V) levels were not affected by GPC1 knockdown (Fig. 5g), suggesting that levels of col(V) containing α3(V) chains, but not col(V) lacking α3(V) chains (for example, α1(V)$_2$α2(V) heterotrimers), are affected by GPC1 interactions.

**α3(V) modulates GPC1 ability to affect proliferation.** WT/PyMT tumour cells infected with Ad-Sh-1 also had markedly decreased proliferation (Fig. 6a), consistent with reported GPC1 abilities to modulate mitogenic responses and stimulate proliferation[23,34]. This marked decrease in proliferative activity was accompanied by significantly decreased Ras signalling and FGFR phosphorylation (Fig. 6b,c and Supplementary Fig. 5), similar to that resulting from α3(V) knockout (Fig. 3g,h), and thus consistent with the possibility that GPC1 and α3(V) interact to affect cell proliferation via the same pathway.

Infection with Ad-Sh-1 of KO/PyMT tumour cells, which already have reduced proliferative potential (above), further reduced proliferation (Fig. 6d), although the difference

in proliferative activity between Ad-Sh-1 and control vector-infected KO/PyMT cells was not as great as the difference in proliferative activity between Ad-Sh-1 and control vector-infected

WT/PyMT cells. Thus, GPC1 contributes to proliferative activity of tumour cells in the presence and absence of the α3(V) chain, although this ability is reduced in the absence of α3(V).

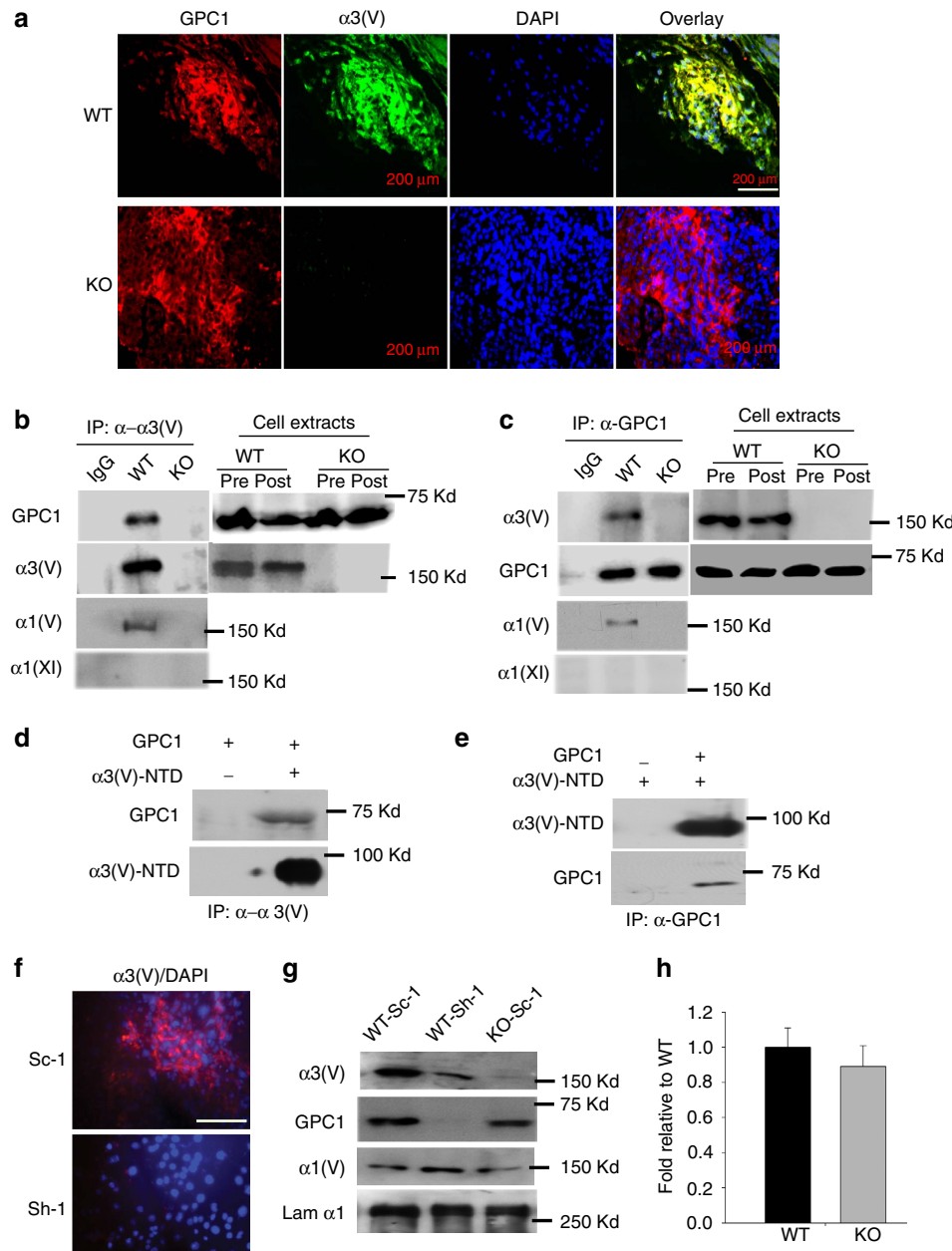

**Figure 5 | GPC1 co-localizes with and binds α3(V) chains in WT/PyMT tumours.** (**a**) Representative immunofluorescent staining for GPC1 (red) and α3(V) chains (green) shows co-localization in WT/PyMT (WT) tumours and the absence of α3(V) staining in KO/PyMT (KO) tumours. Blue staining is DAPI. Overlay panels show areas of GPC1 and α3(V) co-localization (yellow). Immunopreciptiations (IP) from WT/PyMT (WT) or KO/PyMT (KO) tumour extracts were performed using anti-α3(V) (**b**) or anti-GPC1 (**c**) antibodies. Immunoprecipitates were then analysed by immunoblot using antibodies for GPC1, or for α3(V) or α1(V) chains (**b**,**c**). In **b**,**c**, immunoblots are also shown of amounts of α3(V) and GPC1 present in tumour extracts before (Pre) and after (Post) immunoprecipitations of the cognate proteins performed from such extracts, to demonstrate input levels of each protein, and to provide a sense of how much of the available proteins were immunoprecipitated. In **d**,**e**, recombinant GPC1 and α3(V)-NTD sequences were co-incubated, followed by immunoprecipitation with anti-α3(V)-NTD (**d**) or anti-GPC1 antibodies (**e**), and then immunoblotting with anti-GPC1, anti-α3(V)-NTD and anti-α1(V) antibodies. (**f**) Representative immunofluorescent staining for α3(V) and staining for DAPI were performed on WT/PyMT primary tumour cells infected with adenoviral vector Ad-Sh-1 (Sh-1) for shRNA GPC1 knockdown[33], or with scrambled control vector Ad-Sc-1 (Sc-1). (**g**) Immunoblots of ECM associated with WT/PyMT primary tumour cells infected with Ad-Sh-1 or Ad-Sc-1, show lowered α3(V) levels upon GPC1 knockdown. GPC1 knockdown had no effect on α1(V) levels. Laminin α1 chain was stained as a loading control. Immunoblotting of KO/PyMT cells infected with Ad-Sc-1 (KO-Sc-1) served as a control for levels of GPC1 and α1(V) in the absence of α3(V) chains. (**h**) Quantification of three immunoblots from three separate experiments shows no significant difference in GPC1 levels between WT-Sc-1 and KO-Sc-1 cells. White scale bars, 50 μm.

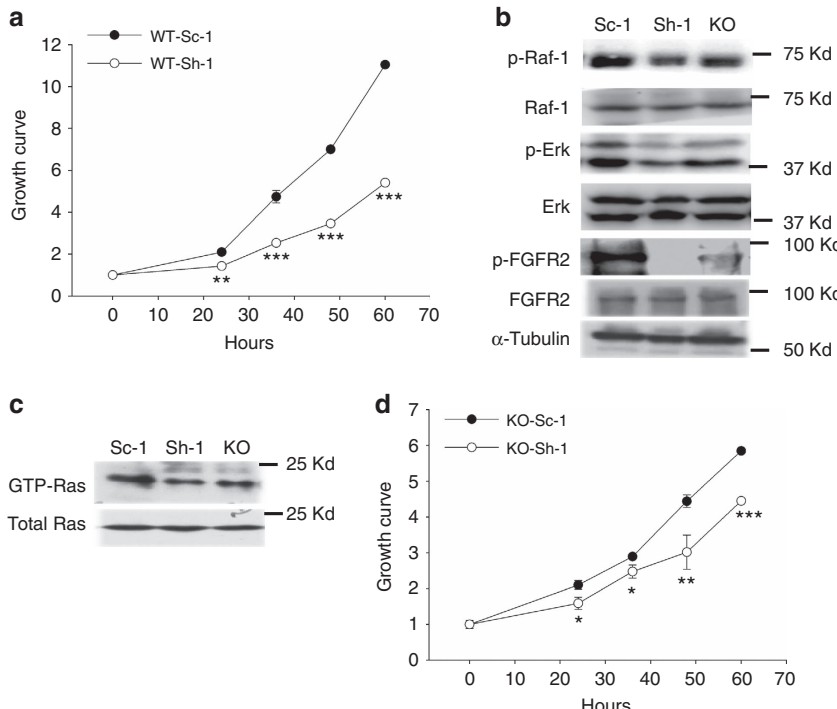

**Figure 6 | Ability of GPC1 to contribute to the proliferative activity of mammary tumour cells is reduced in the absence of the α3(V) collagen chain.** Growth rates, measured via MTT assay, were compared for WT/PyMT (**a**) and KO/PyMT (**d**) primary tumour cells infected with adenoviral vector Ad-Sh-1, for knockdown of GPC1, or with scrambled control vector Ad-Sc-1. All cell proliferation assays were performed in triplicate. Data are presented as mean ± s.d. $P$ values: $* < 0.05$, $** < 0.01$, $*** < 0.005$. Statistical analysis was via two-tailed Student's $t$-test, with differences considered significant at $P < 0.05$. (**b**) Immunoblots are shown of lysates of WT/PyMT primary tumour cells infected with Sc-1 or Sh-1 vector, and of lysates of KO/PyMT (KO) cells. Blots were probed with antibodies to phospho-Raf-1 (p-Raf-1), Raf-1, phospho-Erk1/2 (p-Erk1/2), Erk, phospho-FGFR2 (p-FGFR2), FGFR2, or α-tubulin, the latter as a loading control. (**c**) Lysates were subjected to pull-downs with Raf-1 RBD domain to obtain activated Ras (GTP-Ras), which was then detected by immunoblot, using antibody to Ras. Immunoblots were repeated three times, from three independent tumour or cell lysates. Representative blots are shown. Quantification of results for significance of differences for **b**,**c** is shown in Supplementary Fig. 5.

**Effects of α3(V) ablation on cell growth factor interactions.** GPC1 can affect cell proliferation as a co-receptor for HSPG-binding mitogenic growth factors, but does not affect mitogenic responses to non-HSPG-binding growth factors[23,27]. Thus we determined whether effects of α3(V) ablation on proliferation and Ras/Raf/Erk signalling might operate, at least in part, via affects on ability of GPC1 to act as co-receptor for HSPG-binding growth factors. Towards this end, KO/PyMT and WT/PyMT tumour cells were grown in 10% fetal bovine serum (FBS)-media in the presence/absence of high concentrations of HSPG-binding growth factor FGF2 or non-HSPG-binding growth factor EGF. Interestingly, perhaps because EGF levels in FBS were sufficient for maximal signalling, added EGF had no discernable effect on KO/PyMT or WT/PyMT cell growth (Fig. 7a,b). In contrast, added FGF2 had a large effect in increasing KO/PyMT cell proliferation (Fig. 7a), but a relatively small effect on WT/PyMT cell proliferation (Fig. 7b). Thus, whereas FGF2 concentrations provided by 10% FBS-growth medium seemed sufficient to induce peak signalling/proliferation in WT/PyMT cells, ability of FGF2-augmentatation to increase KO/PyMT proliferation suggests that FGF2 signalling was not at peak levels in 10% FBS-growth medium. Therefore, FGF2 signalling seems less efficient in KO/PyMT than in WT/PyMT cells, and KO/PyMT tumour cells appear to have deficits in HSPG-dependent, but not HSPG-independent, mitogenic signalling. Exogenously added FGF1, which has less affinity for HSPGs than does FGF2 (ref. 35), had no effect on KO/PyMT tumour cell proliferation (Fig. 7a,b).

Interestingly, subsequent to shRNA GPC1 knockdown, augmentation of 10% FBS-media with high levels of FGF2 no longer increased KO/PyMT cell proliferation (Fig. 7c). Thus, the difference in sensitivity to FGF2 between KO/PyMT and WT/PyMT tumour cells, caused by α3(V) ablation, is GPC1-dependent.

To further investigate how α3(V) and GPC1 might interact to affect FGF2 signalling, ELISAs were performed to quantitatively examine FGF2 binding to solid-phase GPC1, α3(V)-NTD, or GPC1 + α3(V)-NTD combined. FGF2 bound either GPC1 or α3(V)-NTD separately, and bound the two combined with somewhat increased affinity (Fig. 7d), with calculated $K_D$ values of 56, 36 and 14 nM, respectively. In complementary ELISAs, GPC1 bound to solid-phase FGF2 or α3(V)-NTD, and bound solid-phase FGF2 + α3(V)-NTD combined with somewhat increased affinity (Fig. 7e), with calculated $K_D$ values of 57, 34 and 9 nM, respectively. In a third set of ELISAs of α3(V)-NTD bound to solid-phase GPC1 or FGF2, and bound GPC1 + FGF2 combined with somewhat increased affinity (Fig. 7f), with calculated $K_D$ values of 33, 35 and 16 nM, respectively. Calculated $K_D$ values of $\sim 57$ nM for FGF2-GPC1 binding in the assays of Fig. 7d,e was similar in magnitude to previous estimates for FGF2 binding to heparin[36], while affinity of FGF2-α3(V)-NTD binding (calculated $K_D$ values of $\sim 35$ nM) in the assays of Fig. 6d,f was a bit higher. Interestingly, each protein bound a mixture of the other two with higher affinity than it bound to either alone (Fig. 7d–f). Consistent with ELISA results, immunoprecipitation of FGF2, α3(V)-NTD or GPC1 co-precipitated each of the other

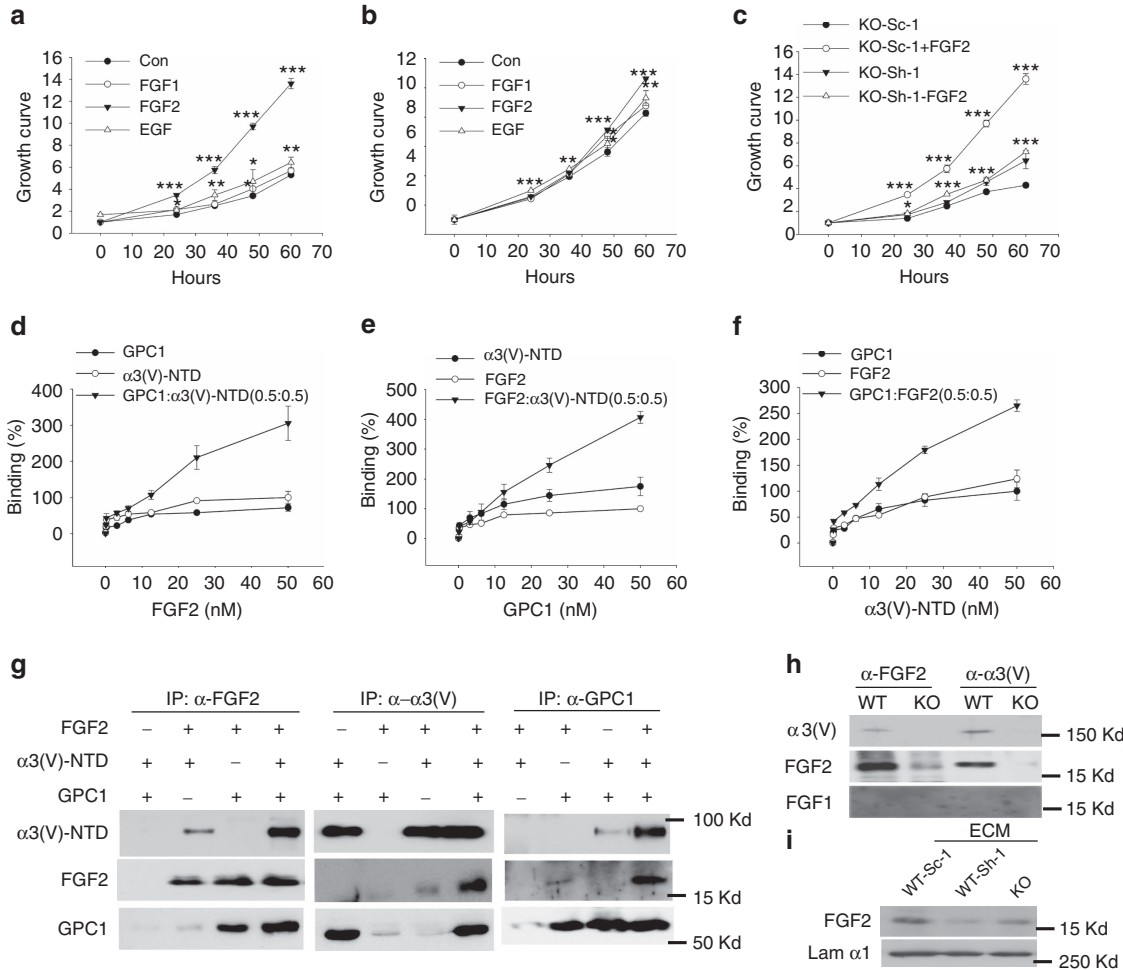

**Figure 7 | α3(V):GPC1:FGF2 interactions modulate FGF2 signalling in tumour cells.** KO/PyMT (**a**) and WT/PyMT (**b**) cells were treated with FGF1, FGF2, or EGF and fold increases in cell numbers at given times, relative to numbers at time 0, were determined using MTT. (**c**) KO/PyMT cells infected with Ad-Sh-1 (Sh-1) for GPC1 knockdown, or scrambled vector Ad-Sc-1, were grown in the presence/absence of FGF2. (**d**) Serial concentrations of FGF2 were incubated in wells coated with GPC1, α3(V)-NTD or both. Plotted values are percent of O.D. obtained from 50 nM FGF2 binding to α3(V)-NTD-coated wells. Serial concentrations of GPC1 (**e**) or α3(V)-NTD (**f**) were incubated in wells coated with FGF2, α3(V)-NTD or both (**e**) or with FGF2, GPC1 or both (**f**). Plotted values are percent of O.D. obtained from incubation of 50 nM GPC1 binding to FGF2-coated wells (**e**) or 50 nM α3(V)-NTD binding to GPC1-coated wells (**f**). Values in **d**-**f** are the mean ± s.d. of three experiments. (**g**) FGF2, α3(V)-NTD and GPC1 were co-incubated, then immunoprecipitated with antibodies to FGF2, α3(V) or GPC1, followed by immunoblotting with the same antibodies. (**h**) Immunoprecipitations from WT/PyMT and KO/PyMT tumour extracts employed antibodies to FGF2 and α3(V), followed by immunoblotting with the same antibodies. (**i**) KO/PyMT or WT/PyMT tumour cells infected with Ad-Sh-1 or scrambled vector Ad-Sc-1 were detached from culture dishes, followed by immunoblotting of ECM remaining on dishes to detect bound FGF2. Laminin α1 immunoblotting was a loading control. All cell proliferation assays and ELISAs were performed in triplicate. Data are presented as mean ± s.d. P values: * < 0.05, ** < 0.01, *** < 0.005. Statistical analysis was via 2-tailed Student's t-test, with differences considered significant at $P < 0.05$.

two proteins; and, in most cases, each protein appeared to bind more strongly to each of the other two in the presence of the third (Fig. 7g). Results thus support the potential for formation of FGF2:α3(V):GPC1 tripartite complexes and suggest some level of cooperative binding. Consistent with inability of FGF1 to affect KO/PyMT cell proliferation (Fig. 7a,b), FGF1 binding to α3(V)-NTD or GPC1 was not detected via immunoprecipitation (Fig. 7h).

In addition to the above *in vitro* interactions with recombinant proteins, immunoprecipitation of endogenous FGF2 from WT/PyMT tumour extracts co-precipitated endogenous α3(V) chains and vice versa (Fig. 7h). Thus, α3(V) and FGF2 are bound together, directly or indirectly, *in vivo*. Consistent with this possibility, α3(V)-free ECM from KO/PyMT tumour cells (Fig. 7i) and ECM of WT/PyMT tumour cells in which GPC1 had been knocked down (Fig. 7i) both contained less FGF2 than did

control WT/PyMT tumour cell ECM. These results are, together, consistent with the possibility that α3(V)-containing ECM and GPC1 act together to concentrate FGF2 at the cell-ECM interface.

**Exogenous α3(V)-NTD induces WT/PyMT tumour cell growth.** Because of our hypothesis that α3(V) chains affect tumour cell growth properties via NTD sequences, we determined whether exogenously added recombinant α3(V)-NTD sequences might affect KO/PyMT tumour cell growth. Interestingly, added α3(V)-NTD increased cultured KO/PyMT tumour cell proliferation (Fig. 8a), accompanied by increased levels of cyclin D, CDK6, MAPK activity, and FGFR phosphorylation; and decreased p-H3$^{ser10}$ levels (Fig. 8b; quantification in Supplementary Fig. 6). Thus, treatment of cells with exogenous α3(V)-NTD is sufficient to rescue some portion of α3(V) function.

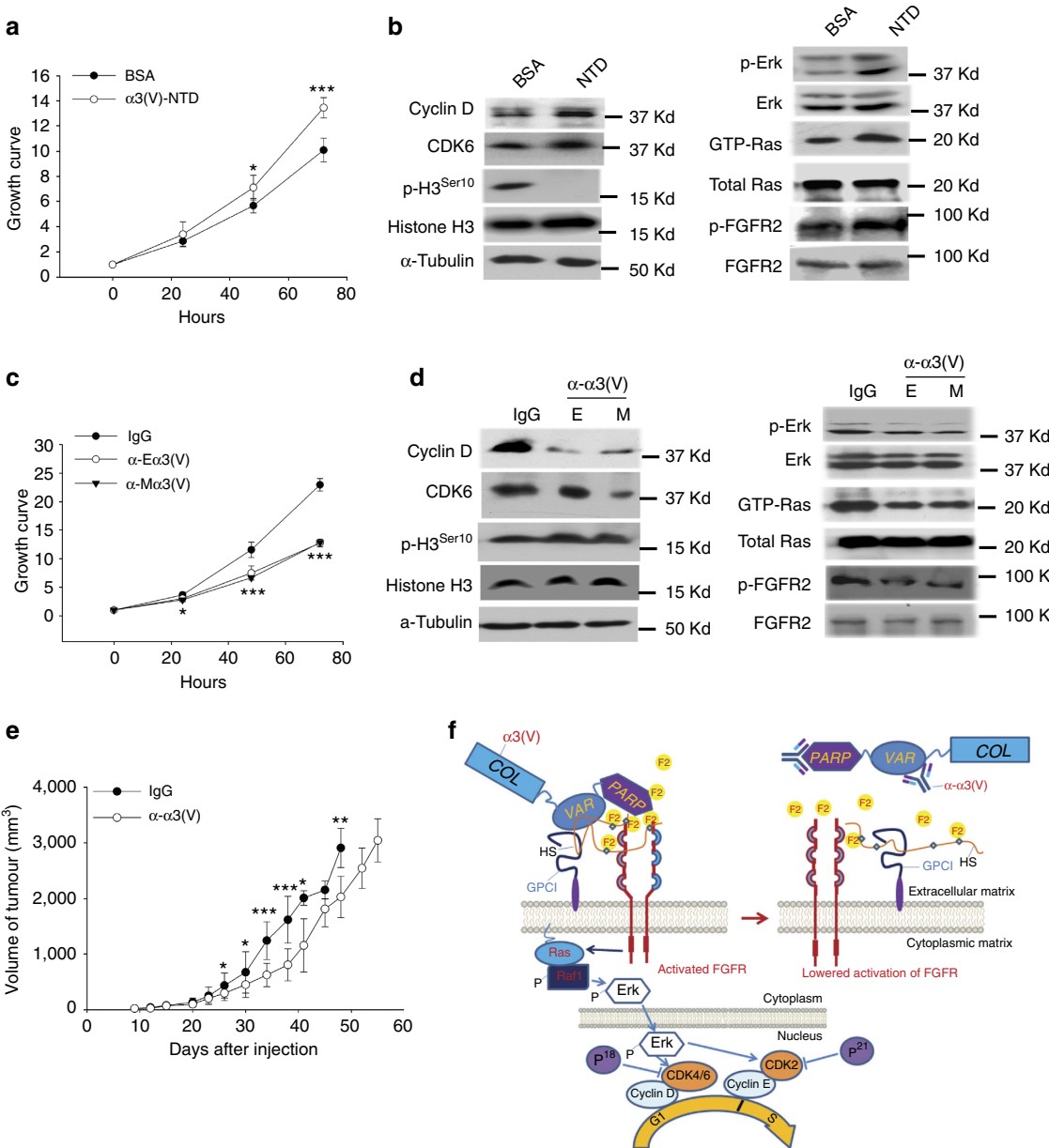

**Figure 8 | Effects of treatment with recombinant α3(V)-NTD sequences and anti-α3(V)-NTD antibodies on *in vitro* and *in vivo* tumour cell growth.** Cultured KO/PyMT cells were treated with recombinant α3(V)-NTD sequences prepared in mammalian cells, followed by monitoring of growth by MTT assay (**a**), or by immunoblotting for cyclin D, CDKs 2 and 6, p-H3$^{Ser10}$, phospho-Erk1/2 (p-Erk1/2), Erk1/2, phospho-FGFR2 (p-FGFR2), and Ras; and for activated GTP-Ras pulled-down from lysate with Raf-1 RBD domain (**b**). Cultured WT/PyMT cells were treated with antibodies raised against recombinant α3(V)-NTD sequences prepared in mammalian cells (M) or *E. coli* (E), followed by monitoring of growth by MTT assay (**c**), or by immunoblotting for cyclin D, CDKs 2 and 6 and p-H3$^{Ser10}$, phospho-Erk1/2 (p-Erk1/2), Erk1/2, phospho-FGFR2 (p-FGFR2), and Ras; and for activated GTP-Ras pulled-down from lysate with Raf-1 RBD domain (**d**). Immunoblotting for α-tubulin was a loading control. (**e**) tumour volumes were measured in C57BL/6 mice injected with WT/PyMT cells and then, 7 days later, with anti-α3(V)-NTD (M) antibodies or with non-immune control IgG. Cell proliferation assays were performed in triplicate. Mice injected with α3(V)-NTD antibody or IgG were n = 8 per group. Data are presented as mean ± s.d. P values: * < 0.05, ** < 0.01, *** < 0.005. Statistical analysis was via two-tailed Student's *t*-test, with differences considered significant at P < 0.05. Immunoblots were repeated three times, from three independent cell lysates. Representative blots are shown. Quantification of results for significance of differences for **b,d** is shown in Supplementary Fig. 6. (**f**) A model in which the acidic PARP and basic variable (VAR) α3(V) subdomains bind basic FGF2 (F2) and acidic GPC1 heparan sulfate (HS) GAG chains, respectively. Via these interactions α3(V) may act as a linker that enhances the efficiency of signalling by HSPG-binding growth factors, which in PyMT cells act via Ras to activate MAPK signalling and the cell cycle.

**Anti-α3(V) antibodies inhibit tumour cell growth.** We next assessed whether treatment with anti-α3(V)-NTD antibodies might affect WT/PyMT tumour cell growth. Such treatment was sufficient to inhibit WT/PyMT tumour cell proliferation in culture (Fig. 8c), with concomitantly reduced cyclin D, CDK 6, MAPK activity, and FGFR phosphorylation levels; and somewhat

increased p-H3$^{ser10}$ levels (Fig. 8d; quantification in Supplementary Fig. 6). To determine whether anti-α3(V)-NTD antibodies could have *in vivo* effects, $5 \times 10^5$ WT/PyMT cells were injected into 10-week-old WT mice, followed 7 days later by injection with 20 μg of anti-α3(V)-NTD antibodies or control IgG twice weekly. The result was significant delay of tumour growth

in antibody-treated mice, with a lesser effect after tumour volume increased past a size threshold (Fig. 8e).

**α3(V) basal and GPC1 luminal localization in human breast**. As GPC1 and α3(V) co-localize and can interact to enhance growth in breast cancer cells (Figs 5 and 7), it was relevant to determine relative distributions of the two proteins in normal mammary duct. As in mouse (Fig. 1b), α3(V) chains were associated with K14-positive basal cells and not with K8-positive luminal cells in normal human mammary glands (Supplementary Fig. 7). However, the inverse was true for GPC1, which was associated with K8-positive luminal cells, and not with K14-positive basal cells of normal human (Fig. 9a) and mouse (Supplementary Fig. 8) mammary glands. Thus, in normal mammary gland, α3(V)-GPC1 interactions would require collaborative basal and luminal cell interaction.

**α3(V) and GPC1 co-localize in human tumours**. Similar to its association with normal K14-positive basal cells, α3(V) chains were associated with the basal-like cells of human triple negative basal-like breast tumours (Fig. 9b). However, as in K8-positive luminal-like[11] tumours of MMTV-PyMT mice (for example, Fig. 1f), α3(V) chains were also associated with human luminal A tumour cells (Fig. 9b). In both human basal-like (Fig. 9b) and luminal A (Fig. 9c) tumours, as in luminal-like[11] MMTV-PyMT tumours (Fig. 5a), α3(V) chains co-localized with GPC1. Interestingly, GPC1 signal seemed consistently higher in luminal A than in basal-like tumours (for example, Fig. 9b,c).

**Correlation of α3(V)/GPC1 expression in breast cancer types**. To determine the extent to which α3(V) and GPC1 expression levels might correlate with human breast tumour phenotypes, we queried the Cancer Genome Atlas Breast Cancer (TCGA_BRCA) dataset[37]. Interestingly, high expression levels for both the α3(V) gene COL5A3 (Fig. 10a) and the GPC1 gene (Fig. 10b) are strongly associated with luminal A breast tumours, with expression level distribution of both genes in human breast cancer in the order luminal A ≫ luminal B ≫ basal-like (P < 0.0001). In fact, association and linear regression analyses demonstrated tight correlation between α3(V) and GPC1 expression levels in these human breast cancer types (Fig. 10c), consistent with the conclusion of meaningful functional interactions between the two proteins. Consistent with the strong association of high α3(V) and GPC1 expression with luminal tumours, high expression of the two proteins also associates strongly with oestrogen receptor-positive (ER+) and progesterone receptor-positive (PR+) tumours, but not with human epidermal growth receptor 2-positive (HER2+) tumours (Fig. 10d,e).

## Discussion

ECM is a key component of microenvironments that affect cell differentiation, function, transformation, tumour growth and metastatic potential[38]. We previously showed the α3(V) chain to markedly affect specialized functions of certain cell types, including adipocytes and pancreatic β cells[6]. Here, we demonstrate α3(V) to affect mammary tumour cell growth in the MMTV-PyMT model, and provide a molecular mechanism for α3(V)-cell interactions. Others have previously shown mammary tumour metastatic potential to be affected by alterations to the density, thickness, orientation and cross-linking of col(I) fibrils[8,38–40]. However, although α1(V)₂α2(V) heterotrimers, the most commonly occurring col(V) form, incorporate into and affect physical properties of interstitial

col(I) fibrils[2–4], α3(V) chains have a pericellular distribution uncharacteristic of col(I), and can be synthesized by cells that do not synthesize col(I), resulting in α3(V)-containing, col(I)-free ECM[6]. In fact, effects of α3(V) ablation on tumour growth are shown here to be predominantly tumour cell-autonomous, due to loss of α3(V) chains normally produced by tumour cells themselves, with only relatively minor non-cell-autonomous effects, presumably contributed by stroma. This is in contrast to the non-cell-autonomous effects on mammary tumour metastasis by col(I) fibrils, exclusively contributed by stroma, consistent with the possibility that α3(V) chains play roles in tumour progression dissimilar to those of col(I).

Known cell surface collagen receptors are the β1-integrins and DDRs (Discoidin Domain Receptors), LAIR-1 (Leukocyte-Associated Immunoglobulin-Like recptor-1), and GPVI (Glycoprotein VI), all of which bind Gly-X-Y repeats of collagen triple helices[41]. In particular, mechanotransduction by β1-integrins appears responsible for changes to tumour cell behaviour in response to changed col(I) fibril physical properties[39,40]. Here, α3(V) chains are shown to interact with mammary tumour cells via a different type of receptor, GPC1, a cell surface HSPG over-expressed in breast and other cancers[23,24,27]. GPC1 is also involved in modulating mitogenic responses to HSPG-binding growth factors and signalling via the Ras/Erk pathway[23–25,27,28] in cancer cells, and in stimulating cell cycle progression in various cell types[34]. That α3(V)-GPC1 interactions are functionally important is supported by the observations that α3(V) ablation results in reduced mammary tumour cell proliferation, Ras/Erk signalling, and cell cycle progression—just those properties stimulated by GPC1. Interestingly, GPC1 knockdown resulted in markedly reduced α3(V) levels in mammary tumour cell-associated ECM. Thus, reciprocal interactions may occur in breast carcinoma, in which tumour cell surface GPC1 contributes to stabilization and/or deposition of α3(V)-containing ECM, while α3(V)-containing ECM enhances tumour cell growth by engaging cell surface GPC1, in a fast-forward process that may affect carcinoma outcomes.

One way in which GPC1 affects cell proliferation is as co-receptor for HSPG-binding mitogenic growth factors[23,27]. Thus, our findings that KO/PyMT tumour cells are deficient in proliferative responses to HSPG-dependent (FGF2), but not HSPG-independent (EGF) growth factors are consistent with the possibility that α3(V) chains aid GPC1 co-receptor function. Moreover, the finding that shRNA knockdown of GPC1 removes ability of FGF2 to rescue KO/PyMT proliferation to WT/PyMT levels (Fig. 7c), indicates the difference in sensitivity to FGF2 signalling between KO/PyMT and WT/PyMT tumour cells to depend on GPC1.

ELISAs and immunoprecipitations with recombinant proteins demonstrated that GPC1 and α3(V)-NTD can bind each other, and that each can bind FGF2. Thus, data were consistent with possible α3(V):GPC1:FGF2 tripartite complex formation, and suggested some level of cooperative binding. Moreover, immunoprecipitations from tumour extracts and analysis of FGF2 amounts associated with the ECM of WT/PyMT, KO/PyMT, and GPC1 knockdown tumour cells supported the conclusion that the three proteins interact in vivo.

Unlike the NTDs of other fibrillar collagen chains, the α3(V)-NTD is not proteolytically trimmed, prompting our previous suggestion that α3(V) might directly interact with cell surfaces via these highly charged sequences[6]. Within the α3(V)-NTD is an acidic (pI = 4.4) 'PARP' subdomain and a basic (pI = 10.3) 'variable' subdomain[7]. Thus, charge interactions between the PARP subdomain and basic FGF2 (pI ≈ 10), and between the variable subdomain and negatively charged GPC1

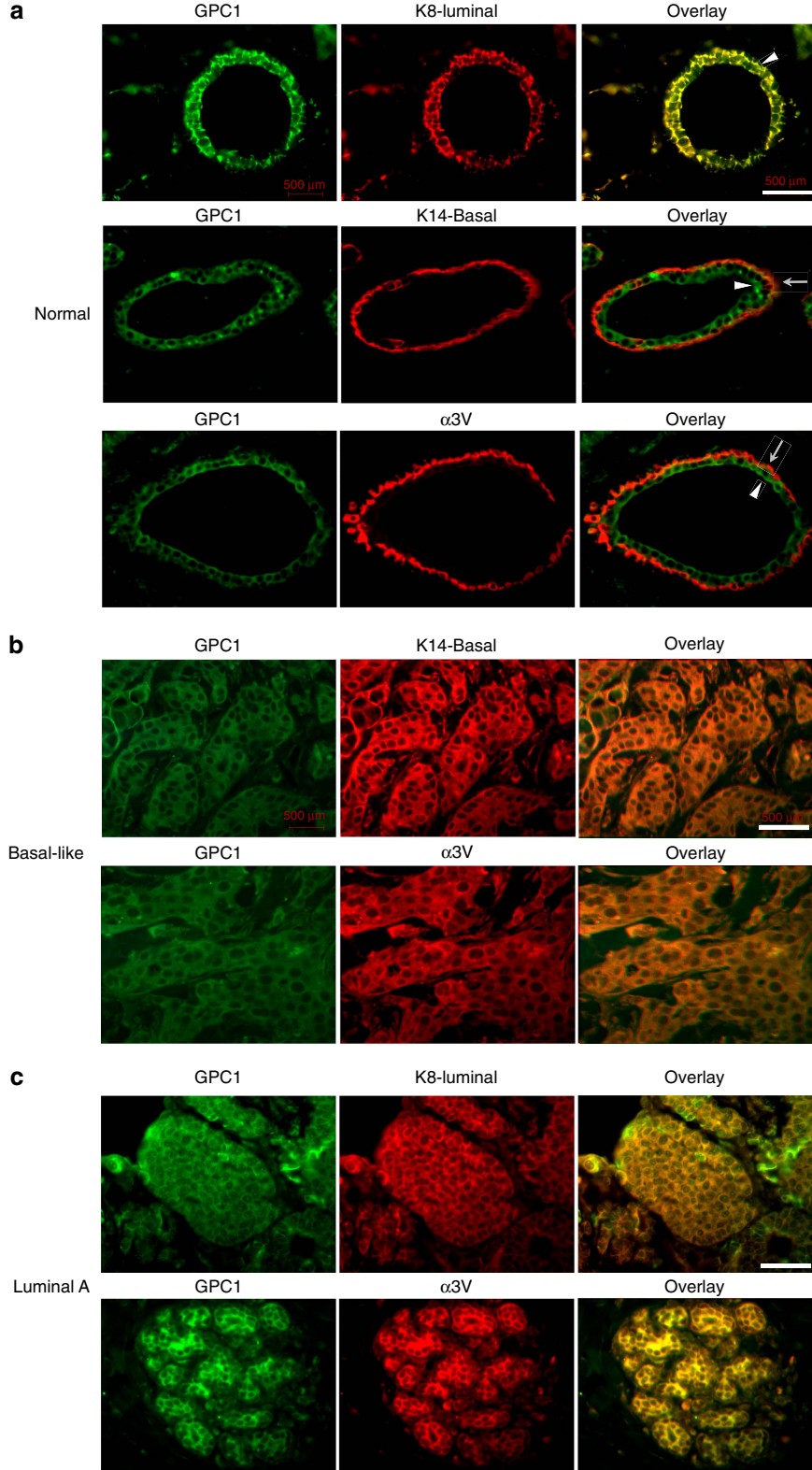

**Figure 9 | α3(V) and GPC1 are associated with basal and luminal cells, respectively, in normal breast, but co-localize in human tumours.** (**a**) Representative immunofluorescence co-localization of GPC1 (green) with marker K8 (red) in luminal cells, but not with marker K14 (red) or with α3(V) (red) in basal cells, in normal human mammary ducts. Arrowheads and arrows denote luminal and basal cells, respectively. (**b,c**) Representative co-localization of GPC1 (green) with basal marker K14 (red) and with α3(V) (red) in human triple negative, basal-like tumours (**b**), and with K8-luminal marker (red) and α3(V) (red) in human luminal A tumours (**c**). Note that GPC1 appeared to be consistently lower in basal-like than in luminal tumours. White scale bars, 50 μm.

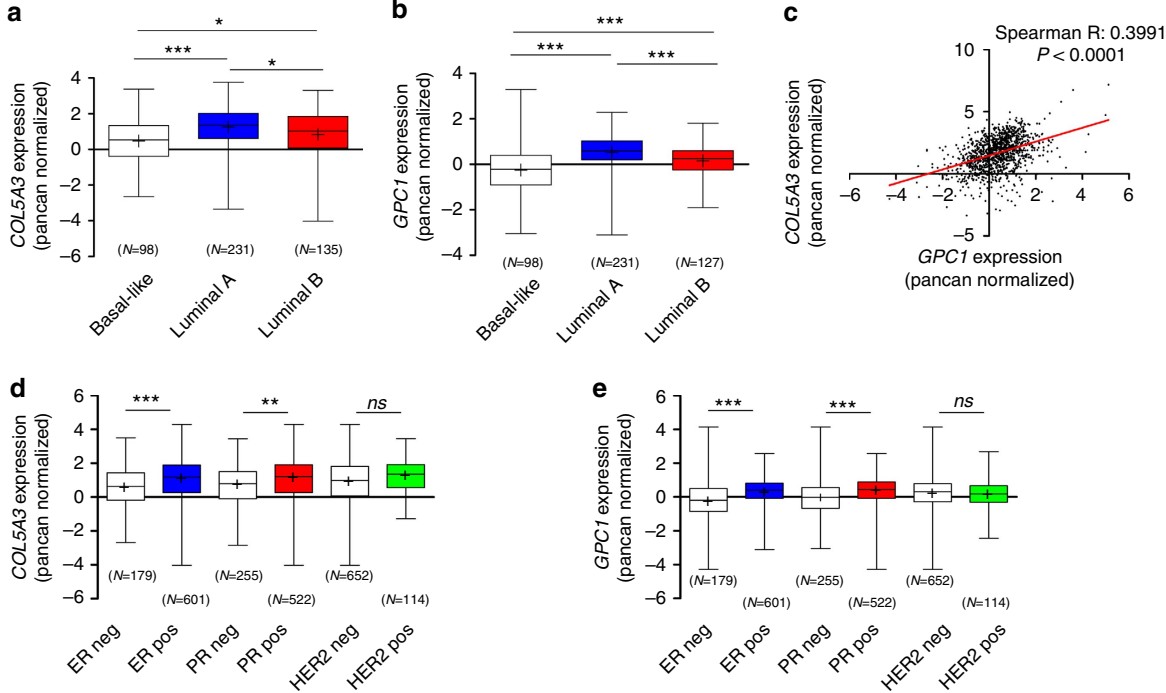

**Figure 10 | Tight correlation of α3(V) and GPC1 expression in human tumour types.** Expression of *COL5A3* (**a,d**) and *GPC1* (**b,e**) (gene expression, Pancan normalized values) in the TCGA Breast Cancer cohort[37], according to (**a,b**) histotype/PAM 50 classification and (**d,e**) hormone/growth factor receptor status. Note markedly higher *COL5A3* and *GPC1* expression in luminal-A type tumours in respect to basal-like and luminal-B types (**a,b**), and association with ER and PR positivity, but not to HER2 expression in **d,e**. All data in **a,b,d,e** are reported as mean (cross mark) ± s.d. The median of each group is also indicated by a continuous line within each boxplot. The number of patients in each group is indicated at the bottom of the corresponding boxplot. (**c**) Association/linear regression analysis demonstrates tight correlation between *COL5A3* and *GPC1* expression in luminal and basal-like tumour types (n = 456 patients, Spearman R, $P < 0.0001$ and R square 0,1740). *$P < 0.05$, **$P < 0.01$, ***$P < 0.001$, as from the Kruskal–Wallis test followed by Dunn's *post hoc* test.

GAG chains could be a straightforward mechanism facilitating α3(V):GPC1:FGF2 complex formation (Fig. 8f). In support of this possibility we found that, unlike FGF2, effects of the acidic FGF1 (pI≈6) on tumour cell proliferation were not affected by α3(V) ablation (Fig. 7a) and that FGF1 did not bind α3(V)-NTD or GPC1 in immunoprecipitations. Thus, lesser FGF1 affinity for HSPGs[35] and for the α3(V)-NTD together likely contribute to reduced potential for α3(V):GPC1:FGF1 complex formation.

Exogenously added α3(V)-NTD sequences significantly enhanced growth of cultured KO/PyMT tumour cells, indicating that α3(V)-NTDs needn't be part of a full-length α3(V) chain or an integral part of the ECM to enhance proliferative behaviour. Thus, future studies employing modified recombinant NTD sequences added to cell cultures, to localize α3(V)-NTD residues involved in effects on tumour cell growth, may further insights into mechanisms involved in α3(V) collagen's effects on cell growth, and further possible effective therapeutic interventions designed to disrupt tumour cell:α3(V) interactions.

In regard to therapeutic interventions, the finding that polyclonal anti-α3(V)-NTD antibodies inhibit mammary tumour cell growth in culture and mammary tumour growth *in vivo* can be extended by generating high-affinity monoclonal antibodies directed against NTD epitopes, to enhance effectiveness in slowing tumour growth. Such preclinical studies may be applied in various models of breast and other cancer types.

In normal mammary gland, α3(V) chains are produced by basal and not luminal cells, whereas GPC1 is produced by luminal and not basal cells. Thus, α3(V)-GPC1 interactions would require collaborative interaction of basal and luminal cells. In contrast, α3(V) and GPC1 co-localize in mouse WT/PyMT, and human

luminal and basal-like, tumours. These results are reminiscent of previous examples in which growth, differentiation, and function in normal breast result from interactions between proteins separately produced by basal and luminal cells, whereas the same proteins are produced by single cells in breast cancer, providing 'gains of autonomy' such that paracrine processes become autocrine and growth advantage is obtained[42]. We speculate that α3(V)-GPC1 co-expression may be an early adaptation that provides growth advantage to transforming luminal epithelial precursor cells, from which both basal-like and luminal tumours are thought to originate[43].

α3(V) and GPC1 expression levels are tightly associated in human breast cancer types, consistent with possible functionally meaningful interactions, with distributions of expression levels of both proteins in the order luminal A ≫ luminal B ≫ basal-like. We speculate that high α3(V)/GPC1 expression levels associated with human luminal A tumours may reflect particular importance of such expression to the growth properties of such tumours. That growth of MMTV-PyMT tumours, which have a luminal tumour gene expression pattern that overlaps those of human luminal tumours[11], is reduced upon α3(V) loss, is consistent with the conclusion that anomalous expression of the normally basal cell-specific α3(V) provides a growth advantage to luminal tumour cells. In contrast, the lesser α3(V)/GPC1 levels associated with basal-like tumours suggest less reliance on an α3(V)-GPC1 functional axis, and more reliance on different, tumour type-specific adaptations[44] for growth advantage.

As α3(V) is expressed in transformed, but not normal luminal cells, it is a potential marker for neoplastic luminal cells. Ability of anti-α3(V) antibodies to slow WT/PyMT tumour growth suggests

α3(V) as a potential target for therapeutic treatments of human tumours in which α3(V)-GPC1 interactions contribute to cell growth.

## Methods

**Mice.** Col5a3[−/−] mice[6] were crossed 10 generations onto a C57BL/6 background. MMTV-PyMT mice on a C57BL/6 background[45] were from Sandra J. Gendler (Mayo Clinic, Scottsdale, AZ, USA). Starting at 8 weeks of age, females were examined for tumours by palpation. For these mice, all palpable tumours were measured for volume calculations. Once detected, primary tumours were measured once weekly with digital calipers. For mice injected with tumour cells, once palpable tumours appeared they were measured twice weekly. The investigator measuring tumours was blinded to the genotype of the mouse hosts. Tumour volumes were calculated with the equation: volume $= a \times b^2 \times 0.52$, in which $a$ is the longest dimension and $b$ the shortest. Mice with a total tumour burden $\geqslant 3000\,mm^3$ were killed.

For metastasis studies, KO/PyMT ($n = 11$) and WT/PyMT ($n = 14$) mice were killed at 20 weeks of age and when total tumour volume reached 10% of body weight ($n = 8$ each), and lungs were harvested after perfusion with 10% neutral buffered formalin. Lungs were subsequently fixed in 10% neutral buffered formalin overnight. Numbers of surface metastasic foci were counted under a dissecting microscope. Lungs were then embedded in paraffin, sectioned and stained with H&E. Areas of tumours and whole lung were measured on sections microscopically to derive the percent of metastatic area.

Mice were housed and treated in accordance with NIH guidelines, using protocols approved by the Research Animal Resources Center and the Institutional Animal Care and Use Committee of the University of Wisconsin-Madison.

**Genotyping.** Genomic DNA from ear punches was analysed for null and WT Col5a3 alleles via PCR employing a primer common for both null and WT alleles (3′-CCATTCCCATTCCCTTGTGAG-5′, reverse) a primer specific for WT sequences that have been deleted in the null allele (5′-GCCTTGTGAATACTGGG CAGC-3′, forward) and a neo-specific primer (5′-TCCTCGTGCTTTACGGTA TCG-3′, forward). Presence or absence of the MMTV-PyMT transgene was ascertained with primers 5′-AGTCACTGCTACTGCACCCAG-3′ (forward) and 3′-CTCTCCTCAGTTCCTCGCTCC-5′ (reverse).

**Human tumour sections.** Totally de-identified, pre-existing formalin-fixed, paraffin-embedded archival sections were obtained from Dr Andreas Friedl of the Department of Pathology and Laboratory Medicine, University of Wisconsin School of Medicine and Public Health. Informed written consent was obtained from the patients. Use of, and all protocols for examination of, these samples were approved by the Heath Sciences Institutional Review Board (IRB) of the University of Wisconsin-Madison.

**Immunofluorescence.** Inguinal mammary glands, tumours and cell cultures were fixed in 4% paraformaldehyde/PBS and tissues were paraffin embedded. For Fig. 1a, 10 μm tissue sections were treated with 1% NaBH₄ to quench autofluorescence. For perilipin staining, sections were blocked with 10% goat serum (Sigma-Aldrich) and incubated with anti-perilipin A/B antibodies (Invitrogen, PA1-1052, 1:500), followed by Alexa Fluor 555 donkey anti-rabbit secondary antibody (Invitrogen, A-31572, 1:1,000). For α3(V) and α-smooth muscle actin staining, antigen retrieval was via autoclaving sections in 10 mM Na-Citrate, pH 4; and sections were treated with 1 mg ml⁻¹ hyaluronidase (type IV-S, Sigma-Aldrich). Sections were then blocked with 5% fish skin gelatin (Sigma-Aldrich) and incubated with anti-α3(V)-NTD (prepared and purified as described below, 1:500) and anti-α-smooth muscle actin (Sigma-Aldrich, clone 1A4, 1:500) antibodies, followed by Alexa Fluor 555 donkey anti-rabbit and Alexa Fluor 488 donkey anti-mouse (Jackson ImmunoResearch, 711-545-152, 1:1,000) secondary antibodies, respectively. For col(V) staining, antigen retrieval was via incubating sections 10 min in 0.5% pepsin/5 mM HCl at 37 °C. Sections were then blocked with 10% FBS and incubated with goat anti-type V collagen antibody (SouthernBiotech, 1350-01, 1:500), followed by Alexa Fluor 488 donkey anti-goat secondary antibody (Jackson ImmunoResearch, 705-545-147, 1:1,000).

For all other immunofluorescence experiments, 10 μm sections or cells were treated with solution H-3300 (Vector Laboratories) for antigen retrieval. Sections were then incubated with anti-α3(V)-NTD and with monoclonal rat anti-mouse Ki-67 (Dako, 49211, 1:100), TROMA-1 monoclonal rat anti-keratin 8 (Developmental Hybridoma Bank, AB-531826, 1:100), polyclonal rabbit anti-keratin 14 (Covance, PRB-155P, 1:100), polyclonal rabbit anti-mouse cleaved caspase 3/Asp175 (Cell Signaling, 9661S, 1:100), p-H3^Ser10 (Cell Signaling, 9701, 1:100), or monoclonal anti-GPC1 (GeneTex, GTX104557, 1:100) antibodies, followed by secondary donkey anti-rat (712-095-153), -rabbit (711-095-152), or -mouse (715-095-151) IgG antibodies conjugated to FITC, or secondary donkey anti-rabbit IgG antibodies conjugated to Cy3 (711-165-152) (all from Jackson ImmunoResearch, 1:1,000). Samples were counterstained with DAPI (Sigma). Fluorescent photomicroscopy was with a Zeiss Axiophot 2 microscope with attached charge-coupled device camera.

**Immunoblotting.** Antibodies to α-tubulin (Millipore, 05-829); Erk (9102), phospho-Erk (9101), Ras (3965), Raf-1 (12552S), p-Raf-1/Ser259 (9421S), FGFR2 (23328S), phospho-FGFR (3471S), H3 (4499), p-H3^Ser10 (9701), cyclins D1, D3, A, B, E (Cyclin Antibody Sampler Kit, 9869T); CDKs 4 and 6 (CDK Antibody Sampler Kit, 9868T); cleaved caspase 3/Asp175 (9661S) (Cell Signaling); GPC1 (GeneTex, GTX104557); GAPDH (SAB2100894), laminin α1 (SAB4501255), FGF2 (SAB2100814), FGF1 (SAB1405808), β-catenin (C2206) (Sigma); α1(XI) (Oncomatrix, IS002); α3(V)-NTD (see below) and α1(V)[46] were all diluted 1:1,000. All were rabbit polyclonal except for CDK4 and 6, and α-tubulin antibodies, which were mouse monoclonal. Uncropped, full scan versions of immunoblots can be found in Supplementary Figs 9–14.

**Measuring active Ras.** Active Ras levels were measured by ability of the Raf-1 RBD (Ras-binding domain) to bind activated Ras-GTP[47]. Tissue or cells were lysed in RIPA buffer (50 mM Tris-HCl (pH 8.0), 150 mM NaCl, 1% NP40, 0.1% SDS, 0.5% DOC, and protease inhibitors (Roche)). After 10 min centrifugation at 12,000 g and 4 °C, supernatant containing 1 μg protein was incubated 2 h with RBD-coupled GST beads (Cytoskeleton, Inc.) at 4 °C. Beads were then centrifuged, washed with RIPA buffer and subjected to SDS–polyacrylamide gel electrophoresis (SDS–PAGE) and immunoblotting.

**Tumour cell culture and injections.** Tumours were dissected and minced under sterile conditions and incubated overnight at 4 °C in growth medium (1:1 F12/DMEM, 10% FBS, 20 ng ml⁻¹ EGF, 0.5 μg ml⁻¹ hydrocortisone, 0.1 μg ml⁻¹ cholera toxin, 10 μg ml⁻¹ insulin, 1× Pen/Strep solution) containing 2 mg ml⁻¹ collagenase type I, 100 units per ml hyaluronidase and 50 μg ml⁻¹ gentamicin. This was followed by 2 h digestion at 37 °C, and centrifugation of cells 10 min at 500 g. Cells were then cultured 3 days in serum-free growth medium, and subsequently in growth medium/5% FBS. In some experiments, KO/PyMT and WT/PyMT cells in growth medium/10% FBS were incubated in the presence of 20 ng ml⁻¹ FGF1 or FGF2, or 10 ng ml⁻¹ EGF (ProSpec). Some tumour cells were infected with adenoviral vectors Ad-Sh-1 or Ad-Sc-1 (ref. 33) (from Andreas Friedl, University of Wisconsin-Madison).

Cell proliferation was measured by colorimetric assay involving mitochondrial dehydrogenase reduction of tetrazolium salt (3-(4,5-dimethylthiazol-2-yl)-2, 5-diphenyltetrazolium bromide; MTT), in which MTT reagent (0.5 mg ml⁻¹, Sigma) was added to each well of cells on a 24 well plate, followed by incubation for 3 h at 37 °C, and then solubilization in isopropyl alcohol (500 μl) and OD readouts at 570 nm. Proliferation was also measured by [³H]-thymidine incorporation, in which cells were preincubated 24 h in serum-free DMEM, which was then replaced with DMEM supplemented with 10% FBS and 0.5 μCi ml⁻¹ [³H]-thymidine. After 24 h, cells were washed with ice-cold PBS, and then treated with ice-cold 10% trichloroacetic acid. After removing the supernatant, cells was washed with cold PBS again, and then dissolved in 0.3 N NaOH. Incorporated radioactivity was measured by liquid scintillation counting.

For tumour cell injection, $5 \times 10^5$ cells in 100 μl sterile PBS were injected into fourth abdominal fat pads, by subcutaneous injection at the base of the nipple of 8-10-week-old mice. Each experimental group was $n = 8$ mice.

**Analysis of cell culture ECM.** Cultured primary tumour cells were washed with ice-cold PBS and then detached from dishes by 5 min treatment with 1% Triton X-100, 150 mM NaCl, 10 mM Tris-HCl, 5 mM MgCl₂, 2 mM EGTA and 0.25 mM DTT, pH 7.2, supplemented with Mini cOmplete protease inhibitor cocktail (Roche). ECM remaining on dishes was washed with the same solution twice, and insoluble material remaining on dishes was extracted with SDS sample buffer/5% 2-mercaptoethanol, and boiled. ECM components were then separated by SDS–PAGE and detected by immunoblotting with laminin α1, α3(V)-NTD, GPC1 and α1(V) antibodies.

**Recombinant α3(V)-NTD and cognate antibodies.** To express murine α3(V)-NTD, nucleotides 64–1089, encoding the α3(V) PARP and variable domains, but lacking signal peptide and COL2 domain sequences, were inserted between NheI and BstBI sites of pcDNA4/TO (Invitrogen) in which a cDNA fragment encoding the BM40 signal peptide had previously been inserted[48]. In the resulting vector, α3(V) NTD sequences are downstream of the BM40 sequences, to optimize secretion, and upstream of pcDNA4/TO sequences that provide a His-tag. Recombinant protein was produced by transfection into 293-TREx cells (Life Technologies) and purified on a HisPur cobalt resin column (Pierce). Anti-α3(V)-NTD antibodies were made against the above protein, but were also, separately, made against His-tagged α3(V)-NTD produced in Escherichia coli from α3(V)-NTD sequences inserted between the NheI and HindIII sites of vector pET28a (Novagen). Anti-α3(V)-NTD polyclonal antibodies raised in rabbits against purified recombinant protein were affinity-purified on a column of the same protein bound to resin.

**ELISAs.** Methods were modified from those of Huang et al.[49]. Briefly, FGF2, GPC1 (R&D Systems), or α3(V)-NTD were separately diluted to 25 nM in buffer A (20 mM Tris-HCl, pH 7.5, 150 mM NaCl, 1 mM ZnCl₂, 1 mM CaCl₂, 1 mM MgCl₂,

pH 7.4) and coated onto 96-well microtitre plates (Costa 3590 high binding, Corning, NY) with 50 µl per well for 16 h at 4 °C. For co-adsorption of GPC1 and α3(V)-NTD, FGF2 and α3(V)-NTD or GPC1 and FGF2 onto wells, each protein was diluted to 12.5 nM in buffer A, so that a final concentration of 25 nM was present in all protein solutions for absorption to wells. Serial concentrations of FGF2, GFPC and α3(V)-NTD incubated in wells coated with binding targets were 50, 25, 12.5, 6.25, 3.125, 1.56 and 0.78 nM. Washing and blocking of wells, dilution of horseradish peroxidase-conjugated secondary antibody, detection with peroxidase substrate and calculations of $K_D$ values were as described[49]. FGF2, GPC1 and α3(V)-NTD antibodies (described above) were diluted to 0.3 µg ml$^{-1}$.

**Co-immunoprecipitation.** For co-immunoprecipitation of recombinant proteins, combinations of 50 ng GPC1, 100 ng α3(V)-NTD and 15 ng FGF2 were pre-incubated 3 h in buffer A (20 mM Tris-HCl, pH 7.5, 150 mM NaCl, 1 mM ZnCl$_2$, 1 mM CaCl$_2$, 1 mM MgCl$_2$, 0.1% Triton X-100 and 1 µg ml$^{-1}$ BSA), followed by antibody addition and 2 h incubation. Protein G or A agarose beads (Roche) were then added, followed by rotation overnight at 4 °C. Beads were then washed with buffer A and subjected to SDS–PAGE for immunoblotting. For immunoprecipitation from cells or tumours, just-confluent primary tumour cell layers or ground tumours were lysed in RIPA buffer, supplemented with Mini cOmplete protease inhibitor cocktail. Lysed samples were centrifuged 10 min at 12,000 g after sonication, and supernatants were subjected to immunoprecipitation, as described above for recombinant proteins. Extracts were 0.3 mg protein ml$^{-1}$. Antibodies were added to 900 µl of extracts for pull-downs. For Fig. 5b,c, 10 µl of extract, either before or after the immunoprecipitations performed on these extracts, were immunoblotted to demonstrate input levels of each protein and provide a sense of how much of the available proteins were brought down in the immunoprecipitation experiments. Antibodies for immunoprecipitation were as described above except for monoclonal anti-GPC1 antibody (Santa Cruz, SC-101827).

**Flow cytometry.** For DNA profile analysis, primary tumour cells were fixed with 70% ethanol at 4 °C and then incubated with 250 µg ml$^{-1}$ RNase A and 10 µg ml$^{-1}$ propidium iodide (invitrogen) at 37 °C for 30 min. FACS was via FACSCalibur flow cytometer (Becton Dickinson) and data were analysed with FlowJo software. For DNA staining, live cells were suspended in culture medium (10$^6$ cell ml$^{-1}$) after trypsinization. Hoechst 33342 was then added (2 µg ml$^{-1}$) followed by 30 min incubation at 37 °C. The flow cytometer (Arial II with biosafety cabinet) was set for excitation at 340–380 nm, for sorting G1, S and G2/M phase cells, after which cells were released into culture medium. Cultured cells were harvested for immunoblotting or fixed for DNA profiling at intervals of 3–6 h.

For cell synchronization by double thymidine block, 30% confluent cells were treated with 2 µM thymidine. After 18 h, thymidine was removed by washing with PBS, and cells were then incubated 9 h in fresh media. Subsequently, 2 µM thymidine was again added, followed by 15 h incubation. Cells were then washed with PBS, released into fresh medium, and harvested at indicated time points for immunoblotting.

**Antibody treatment of mice and tumour cells.** Primary tumour cell proliferation was quantitated by MTT assay at 24, 48 and 72 h after adding 0.5 µg ml$^{-1}$ anti-α3(V)-NTD antibody to culture medium. For immunoblotting, cells treated 48 h with 0.5 µg ml$^{-1}$ antibody were lysed with SDS loading buffer. For *in vivo* immunotherapy, $5 \times 10^5$ WT/PyMT cells were introduced into fourth abdominal fat pads of 10-week-old female C57BL/6 mice by subcutaneous injection. Seven days later, mice were divided into a treated group (n = 8), receiving 20 µg α3(V)-NTD antibody, or untreated group (n = 8), receiving 20 µg IgG, subcutaneously at the abdominal midline, twice weekly. Tumour sizes were measured twice weekly.

**Statistics.** Most analyses, other than TCGA database analysis, were performed with two-tailed Student's t-test, with differences considered significant at $P < 0.05$. Variances were in each case similar between the two groups being statistically compared. Survival analyses were performed using the Kaplan–Meier method. Survival in groups was compared using the Log-Rank test. Kaplan–Meier survival plots were generated using MedCalc (version 11.1). For TCGA database analysis, the Cancer Genome Atlas Breast Cancer (TCGA_BRCA) data set was accessed and the following parameters downloaded for further analysis: IlluminaHiSeq (pancan normalized) for gene expression, HER2 final status, ER status and PR status[37] for hormone receptors' classification, and PAM50 subtype[37] for histological clustering. Normality of data distribution was verified by D'Agostino-Pearson omnibus test, and non-parametric 1-way analysis of variance (Kruskal–Wallis) followed by Dunn's test, performed accordingly. A P value < 0.05 was considered as significant.

**Data availability.** All the relevant data are available within the Article and Supplementary Files, or available from the authors upon request.

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

## Acknowledgements

We thank Dr Andreas Friedl for the gift of shRNA plasmids and tumour sections and Dr Sandra J. Gendler for the gift of MMTV-PyMT mice. We also thank the University of Wisconsin Carbone Cancer Center (UWCCC) for subsidized (NIH/NCI P30CA014520—UWCCC support grant) use of Shared Resources. This work was supported by a Centre of Excellence Grant 284606 from the Academy of Finland (V.I.) and by NIH grants R01-AR47746 and PO1AI084853 (D.S.G.).

## Author contributions

D.S.G. designed and supervised research studies, analysed the data and wrote the manuscript. G.H. designed research studies, conducted experiments, acquired and analysed the data. G.G. conducted an experiment, acquired and analysed the data. V.I. performed the statistical analyses of the TCGA databases.

## Additional information

**Competing financial interests:** The authors declare no competing financial interests.

