## [Peer Review File · Nature Communications]

Reviewers' Comments:

Reviewer #1 (Remarks to the Author)

This manuscript details a decrease in tumor formation in mice in which the alpha-3 chain of collagen V has been knocked out. Col a3(V) is a unique collagen subtype that is not well characterized, and has not been linked to tumor progression, and thus is a novel finding. The knock out leads to decreased Ras/MAPK signaling, and delayed cell cycle progression.

The authors demonstrate that glypican-1 (GPC1) is a likely cell-surface receptor for Col a3(V) by co-immunostaining and co-immunoprecipitation. The interaction site was localized to the highly charged sequences of the N-terminal domain (NTD) of Col a3(V). Importantly, tumor growth is inhibited by injection of the Col3a(V) NTD, which shows potential therapeutic relevance.

The underlying hypothesis that the expression of Col a3(V) and one of its receptors, glypican-1, in the same cell contributes to a dis-regulated growth state is a fundamental shift in the way we view matricellular proteins, their receptors and co-receptors.

While interesting, the work in its present form still requires some additions:

- 1) The functional link between Cola3(V) and GPC1 needs to be further strengthened. The use of NTD is a start, but there could be effects on other Cola3(V) receptors that also bind the highly charged domain. The role of GPC1 in this model would benefit from using knock-down cells in vivo (as mammary fat pad injection) to demonstrate a functional role of GPC1 in tumor formation.
- 2) Throughout the manuscript, the data shown are mostly an example of one IF image or one western blot. Quantitation of the data from the western blots, and of the co-localization for IF images is needed throughout.
- 3) All images need scale bars. Moreover, often it is difficult to determine if similar regions of tissue are being compared. A representative H&E image of the tumors would help. Arrows would help guide the reader's attention in the IF figures.
- 4) For co-IP results in Figure 5, it is necessary to show the total cell lysate blotted, so that it is possible to get a sense of how much of the available protein was co-immunoprecipitated.

Reviewer #2 (Remarks to the Author)

Huang et. al

A3(V) collagen regulates breast tumor growth via glypican-1-mediated effects

This manuscript demonstrates that loss of a3(V) collagen (Col5a3) inhibits mammary tumor progression via cell autonomous regulation of proteoglycan glypican (GPC1) and coreceptor FGF2. Authors show that Col5a3 KO mice when crossed to PyMT breast cancer model slowed tumor volume and increased survival of mice. Authors show that a3(V) contributes to cancer growth in PyMT model as both cell autonomous and non-cell autonomous with cell autonomous having a much greater contribution. Authors show that KO/PyMT tumors have lower proliferation compared to control mice and in cells they have reduced Ras-mediated signaling and delayed cell cycle progression with blocks at G1 and G2/M compared to control cells. Authors show that a3(V) interacts with GPC1 in PyMT tumors and this interaction is lost in KO/PyMT cells compared to controls. Authors show that effects of loss of a3(V) is associated with binding of FGF2 to GPC1. Lastly, authors show treating PyMT tumor cells in vitro and in vivo with blocking a3(V) antibody inhibits growth and that a3(V) and GPC1 expression are tightly correlated in luminal A cancers.

Overall, the manuscript provides some novel data regarding $\alpha 3(V)$ collagen (Col5a3) as regulator of breast cancer and possible therapeutic target. This manuscript is well written and contains strong data to support major conclusions. However, some issues need to be addressed before publication.

Major issues:

1. Authors show that $\alpha 3(V)$ contributes to cancer growth in PyMT model as both cell autonomous and non-cell autonomous functions. Figures 3A, 5A should be done using KO/PyMT cells in wildtype mice to ensure effects seen are cell autonomous.
2. Figures 5b-e needs to show levels of GPC1 in whole cell extracts (input) to ensure KO/PyMT cells don't have altered GPC1 levels (Fig. 5G looks like less GPC1 in KO cells).
3. Based on the author's model, KO/PyMT cells would be expected to have much less FGFR phosphorylation in vivo and in vitro. Authors should examine this to confirm model. Should also examine FGFR phosphorylation, Ras/Erk pathway in Figure 7.
4. Connection that $\alpha 3(V)$ regulates PyMT cell growth and signaling via GPC1 is rather weak. Authors should use their GPC1 RNAi in wildtype PyMT cells to test whether reducing GPC1 inhibits growth, Ras-signaling and FGFR phosphorylation.

Minor Issue:

1. Need to show stats for Figure 1D (authors mention in text they see significant increase but do not show stats).
2. Page 7 Line 146, 148, 150 incorrectly refers to Fig. 3.
3. Need to show total Raf-1 for Fig. 3e.
4. Need to show total levels of Histone H3 for blots in Fig. 4f, 4G.

Response to Referees:

Reviewer #1 (Remarks to the Author):

This manuscript details a decrease in tumor formation in mice in which the alpha-3 chain of collagen V has been knocked out. Col a3(V) is a unique collagen subtype that is not well characterized, and has not been linked to tumor progression, and thus is a novel finding. The knock out leads to decreased Ras/MAPK signaling, and delayed cell cycle progression.

The authors demonstrate that glypican-1 (GPC1) is a likely cell-surface receptor for Col a3(V) by co-immunostaining and co-immunoprecipitation. The interaction site was localized to the highly charged sequences of the N-terminal domain (NTD) of Col a3(V). Importantly, tumor growth is inhibited by injection of the Col3a(V) NTD, which shows potential therapeutic relevance.

The underlying hypothesis that the expression of Col a3(V) and one of its receptors, glypican-1, in the same cell contributes to a dis-regulated growth state is a fundamental shift in the way we view matricellular proteins, their receptors and co-receptors.

While interesting, the work in its present form still requires some additions:

1) The functional link between Cola3(V) and GPC1 needs to be further strengthened. The use of NTD is a start, but there could be effects on other Cola3(V) receptors that also bind the highly charged domain. The role of GPC1 in this model would benefit from using knock-down cells *in vivo* (as mammary fat pad injection) to demonstrate a functional role of GPC1 in tumor formation.

Response – In response, please note that various studies by others have previously provided strong evidence of the role of GPC1 in the growth of tumor cells, *in vitro* and *in vivo* (and to the mitogenic effects of heparin-binding growth factors) (e.g. Kleeff et al *J Clin Invest* 102, 1662, 1998; Kleeff et al, *Pancreas* 19, 281, 1999; Matsuda et al, *Cancer Res* 61, 5562,

2001; Ding et al, *J Cell Biol* 171, 729, 2005; Su et al, *Am J Pathol* 168, 2014, 2006; etc). To emphasize these previously derived insights into the roles of GPC1 in tumor growth and cell proliferation, we've added the above references to the revised manuscript, and cite them appropriately in the text. Because of these previous insights into the importance of GPC1 to mitogenic effects and tumor growth, we do not think that it would be a large contribution by the current study to demonstrate similar GPC1 effects in the MMTV-PyMT system. Another consideration is that our adenoviral shRNA vectors, used in the paper for *in vitro* studies, would not be appropriate for the suggested *in vivo* studies, as GPC1 knockdown with these vectors would not persist long enough for the *in vivo* time course. We could obtain lentiviral vectors for a more stable GPC1 shRNA knockdown that would be more suitable for the suggested *in vivo* experiments. However, we estimate that such experiments, especially given the expected slow growth rates of GPC1-knockdown tumors, would take ~5 months. This would preclude our being able to submit the revised manuscript within the permitted three months limit allowed by *Nature Comm*. Given the above considerations, we ask that, instead of the suggested *in vivo* experiments, the Reviewer see our response to major point 4 of Reviewer 2, which overlaps this first point of Reviewer 1. In our response, we have added a new Fig. 6, for the revised manuscript, in which cell culture data show that GPC1 knockdown in WT/PyMT tumor cells markedly decreases tumor cell growth, accompanied by decreased Ras signaling and FGFR phosphorylation, effects similar to those resulting from $\alpha 3(V)$ knockout, and thus consistent with the possibility that GPC1 and $\alpha 3(V)$ interact to affect proliferation via the same pathway.

Please note that adding the new Figure 6 to the revised manuscript necessitated moving Table 1 from the manuscript to Supplemental information, as Supplementary Table 1, as otherwise the manuscript would have been over the 10 display item limit.

2) Throughout the manuscript, the data shown are mostly an example of one IF image or one western blot. Quantitation of the data from the western blots, and of the co-localization for IF images is needed throughout.

Response - In response, please note that immunoblots for each experiment were repeated 3 times and that the immunoblots shown in figures are representative of the three. We now note this in the legends for figures 3, 4, 6, and 8 and quantify the data from sets of 3 blots represented in these figures to bolster important conclusions on differences in MAPK signaling and FGFR phosphorylation; and differences in cell cycling. The quantifications are included in three new and one revised supplementary figures in Supplemental information (Supplementary Figures 3 – 6) and are referred to in the legends of Figures 3, 4, 6, and 8, and in the Results text, or the manuscript. As an example of how these revisions were handled, here is what was added to the legend for the new figure 6 in the revised manuscript:

“Immunoblots were repeated 3 times, from 3 independent tumor or cell lysates. Representative blots are shown. Quantification of results for significance of differences for panels **b** and **c** is shown in Supplementary Fig. 5.” Reference in the corresponding text was to not only the data of the new Fig. 6, but also to the quantification of these data in Supplementary Fig. 5.

For IF co-localization images, each image shown was representative of multiple observed fields. In recognition of this, we've changed "immunofluorescence staining" to "representative immunofluorescence staining" and changed "immunofluorescence co-localization" to "Representative immunoco-localization" in figures 1, 5, and 9. Quantification did not seem necessary for these IF images, as they dealt with localization and co-localization of signals for various proteins, rather than comparison of signal/protein levels. The exceptions were the IF images of Fig. 3, in which quantification had already been performed in the original manuscript.

3) All images need scale bars. Moreover, often it is difficult to determine if similar regions of tissue are being compared. A representative H&E image of the tumors would help. Arrows would help guide the reader's attention in the IF figures.

Response - We apologize for the omission of scale bars, and have added scale bars to figs. 1a, b, c, f and g; 3a; 5a and f; and 8, and describe such scale bars (all are 50 μ m) in the revised figure legends. In addition, we've added arrows and arrowheads to Fig. 1b and c, Fig. 8a, and Supplementary Fig. 5 – to indicate basal and luminal cells, respectively. Arrows and arrowheads also now denote basal and luminal cells in an untransformed duct in the tumor in Fig. 1f, and an arrow now marks the basal region of an untransformed duct in a tumor in Fig. 1g. Arrows and arrowheads did not seem appropriate for other IF images in the figures, in which they would not have been useful in pointing out morphological features, and in which different colors make the identity of different stained proteins self-evident. The reviewer also suggested "representative H&E images of the tumors". However, sections adjacent to those used for the IF images in the paper, are not available for H&E staining. As we do not believe that such H&E stained sections would uncover morphological features that would change or augment any of the conclusions drawn from the IF images, we request that such H&E stained sections not be required.

4) For co-IP results in Figure 5, it is necessary to show the total cell lysate blotted, so that it is possible to get a sense of how much of the available protein was co-immunoprecipitated.

Response - In response, we've repeated α 3(V) and GPC1 pull downs for Fig. 5b and c, and now also include blots of the corresponding total tumor lysates, both prior to and post immunoprecipitations performed from such extracts, to demonstrate input levels of each protein, and provide a sense of how much of the available proteins were immunoprecipitated. The figure legend has been changed accordingly, as has the corresponding section of Methods, which now describes the volume of extract used for immunoprecipitations and the volumes of extract loaded onto gels for immunoblots of total extract/protein input.

Reviewer #2 (Remarks to the Author):

Huang et. al

A3(V) collagen regulates breast tumor growth via glypican-1-mediated effects

This manuscript demonstrates that loss of $\alpha 3(V)$ collagen (Col5a3) inhibits mammary tumor progression via cell autonomous regulation of proteoglycan glypican (GPC1) and coreceptor FGF2. Authors show that Col5a3 KO mice when crossed to PyMT breast cancer model slowed tumor volume and increased survival of mice. Authors show that $\alpha 3(V)$ contributes to cancer growth in PyMT model as both cell autonomous and non-cell autonomous with cell autonomous having a much greater contribution. Authors show that KO/PyMT tumors have lower proliferation compared to control mice and in cells they have reduced Ras-mediated signaling and delayed cell cycle progression with blocks at G1 and G2/M compared to control cells. Authors show that $\alpha 3(V)$ interacts with GPC1 in PyMT tumors and this interaction is lost in KO/PyMT cells compared to controls. Authors show that effects of loss of $\alpha 3(V)$ is associated with binding of FGF2 to GPC1. Lastly, authors show treating PyMT tumor cells in vitro and in vivo with blocking $\alpha 3(V)$ antibody inhibits growth and that $\alpha 3(V)$ and GPC1 expression are tightly correlated in luminal A cancers.

Overall, the manuscript provides some novel data regarding $\alpha 3(V)$ collagen (Col5a3) as regulator of breast cancer and possible therapeutic target. This manuscript is well written and contains strong data to support major conclusions. However, some issues need to be addressed before publication.

Major issues:

1. Authors show that $\alpha 3(V)$ contributes to cancer growth in PyMT model as both cell autonomous and non-cell autonomous functions. Figures 3A, 5A should be done using KO/PyMT cells in wildtype mice to ensure effects seen are cell autonomous.

Response - We believe that the Reviewer was referring to Figs. 3A and 3B, as these related panels both deal with effects of $\alpha 3(V)$ on cancer growth, whereas Fig. 5A does not. In response to the Reviewer's comment we've added Ki-67 staining (new panel c), and quantification of such staining (new panel d), of WT/PyMT and KO/PyMT tumor cells injected into WT C57BL/6 mice, to show that the effects of $\alpha 3(V)$ on Ki-67 indices are indeed cell-autonomous. The figure legend and text have been revised accordingly.

2. Figures 5b-e needs to show levels of GPC1 in whole cell extracts (input) to ensure KO/PyMT cells don't have altered GPC1 levels (Fig. 5G looks like less GPC1 in KO cells).

Response - In response, and in response to a similar request by Reviewer 1 (above) we've repeated the immunoprecipitations of GPC1 and $\alpha 3(V)$ for Fig. 5b and c, and now include, for each pull-down experiment, blots of the levels of these proteins in corresponding total tumor lysates, both prior to and post immunoprecipitations performed from such extracts, to demonstrate input levels of both proteins, and to provide a sense of how much of available proteins were immunoprecipitated. The figure legend has been changed accordingly, as has the corresponding section of Methods, which now describes how much extract was used for

immunoprecipitations and how much extract was loaded on gels for immunoblots. Fig. 5d and e were not repeated, as these involved purified recombinant proteins, and not cell/tumor extracts.

3. Based on the author's model, KO/PyMT cells would be expected to have much less FGFR phosphorylation *in vivo* and *in vitro*. Authors should examine this to confirm model. Should also examine FGFR phosphorylation, Ras/Erk pathway in Figure 7.

Response - In response, we've added immunoblots for total FGFR and p-FGFR to Fig. 3g, which indeed show decreased FGFR phosphorylation in KO/PyMT tumor cells *in vivo* and *in vitro*. We've also added Ras/Erk-signaling (total Ras, GTP-Ras; total Erk1/2, p-Erk1/2) and FGFR phosphorylation (total and p-FGFR) immunoblots to Fig. 8 (this was Fig. 7 in the original manuscript, but has been renumbered subsequent to the newly added Fig. 6 – see below), as requested by the reviewer. The new data bolster conclusions based on the previous blots.

4. Connection that $\alpha 3(V)$ regulates PyMT cell growth and signaling via GPC1 is rather weak. Authors should use their GPC1 RNAi in wildtype PyMT cells to test whether reducing GPC1 inhibits growth, Ras-signaling and FGFR phosphorylation.

Response – Please note that in the original manuscript we did show that shRNA knockdown of GPC1 in WT/PyMT cells profoundly inhibits cell growth (original Supplementary Fig. 4). In response, to the Reviewer's point, we've added a new Fig. 6, to the revised manuscript, that contains these data (which may have been obscured in the original Supplementary Fig. 4), and to which we've also added new data showing that the inhibition of cell growth resulting from GPC1 shRNA knockdown is accompanied by decreased Ras-signaling (Raf-1 and Erk1/2 phosphorylation, and activation of Ras to GTP-Ras), and FGFR phosphorylation. In the revised text of the manuscript (pages 12 and 13) we describe these results as follows (with new/added text in bold): “WT/PyMT tumor cells infected with Ad-Sh-1 also had markedly decreased proliferative activity (**Fig. 6a**), consistent with reported abilities of GPC1 to modulate mitogenic responses and stimulate proliferation^{24 30}. **This marked decrease in proliferative activity was accompanied by decreased Ras signaling and FGFR phosphorylation (Fig. 4 b and c), similar to that resulting from $\alpha 3(V)$ knockout (Fig. 3 g and h), and thus consistent with the possibility that GPC1 and $\alpha 3(V)$ interact to affect cell proliferation via the same pathway.**”

Minor Issue:

1. Need to show stats for Figure 1D (authors mention in text they see significant increase but do not show stats).

Response - Our previous graphing of the survival data for Figs. 1d and 2a and c was with a version of SigmaPlot that did not allow statistical treatments of the Kaplan-Meier (KM) plots. In response to the Reviewer's comments, we re-analyzed the data for these figures, using MedCalc, version 11.1. The original KM plots have been replaced with the new plots in the

revised Figs. 1d, 2a, and 2c, and statistical treatment shows the differences in survival ratios to have been significant for KO/PyMT vs WT/PyMT mice in Fig. 1d ($P < 0.0001$), and for WT vs *Col5a3*^{-/-} mice injected with WT/PyMT tumor cells ($P < 0.0001$) in Figs. 2a and 2c. In addition, in Fig. 2c there was found to be no significant difference in survival times between WT vs *Col5a3*^{-/-} mice injected with KO/PyMT tumor cells ($P < 0.98$). Thus, results of 2a and 2c are further shown to be consistent with our previous conclusion that tumor cell-autonomous effects of *Col5a3* ablation have much more of an effect on survival and tumor progression than do non-cell-autonomous effects. *P* values have been added to the figure legends and/or text (together with explanations, where appropriate). Description of the statistical treatment has been added to Methods.

2. Page 7 Line 146, 148, 150 incorrectly refers to Fig. 3.

Response - We are thankful to the Reviewer for noticing textual errors on page 7 in references to Fig. 3. We also found similar errors at the bottom of original page 6 and the top of original page 8. In response, we've changed "... Ki-67 than do WT/PyMR tumors (Fig 3a)" (bottom, original page 6), To "... Ki-67 than do WT/PyMR tumors (Fig 3 a **and b**)"; "...MTT (Fig 3b) or ³H-thymidine incorporation (Fig 3c)" (original page 7, line 5,) To "...MTT (Fig 3c) or ³H-thymidine incorporation (Fig 3d)"; "Ras levels did not differ (Fig 3 d and e)..." (original page 7, line 5 from bottom) To "Ras levels did not differ (Fig 3 e and f)..." ; "...extracts (Fig 3e), as were levels of activated phosphor-Raf-1 (Fig 3d)" (original page 7, line 5 from bottom) To "...extracts (Fig 3f), as were levels of activated phosphor-Raf-1 (Fig 3e)"; "...samples (Fig 3d), consistent..." (bottom line, original page 7) To "...samples (Fig 3e), consistent..."; and "...found no differences (Fig 3d)" (line 5, original page 8) To "...found no differences (Fig 3e)".

3. Need to show total Raf-1 for Fig. 3e.

Response -As called for by the Reviewer, we've added immunoblotting for total Raf-1 to Fig. 3e.

4. Need to show total levels of Histone H3 for blots in Fig. 4f, 4G.

Response - As called for by the Reviewer, we've also added immunoblotting for total H3 to Fig. 4f. In addition, although the Reviewer also asked for adding H3 to the blots of Fig. 4g, the latter panel is a histogram of quantification of immunofluorescence staining. We believe that the Reviewer was instead referring to Fig. 4h, and we've thus added staining for H3 to the blots of Fig. 4h.

In conclusion, we've attempted to respond to the reviewers' comments in what we feel to be a reasonable way, and we hope that our responses are deemed suitable. We are thankful for suggestions from the Reviewers that we feel have improved the manuscript and made it even stronger.

REVIEWERS' COMMENTS:

Reviewer #1 (Remarks to the Author):

In this revised manuscript, Huang et al add important new data (Figure 6) that strengthens their central mechanism linking Col a3(V) to GPC-1. While this reviewer requested in vivo experiments, the authors make appropriate argument why the feasibility and value of such experiments makes them beyond the scope of this paper. By strengthening the in vitro data the authors have sufficiently addressed my concern.

Throughout the manuscript, there has been more careful attention to details such as scale bars, and the number of replicates has been indicated. Quantitation of important blots has been added to the supplementary material.

I now find this paper suitable for Nature Communications. The novel findings regarding the role of Col a3(V) in tumor growth, and the elucidation of the GPC-1 and FGF2 mechanisms will be of interest in the field at large.

Reviewer #2 (Remarks to the Author):

Authors have addressed all major and minor issues and thus manuscript is now acceptable for publication.